# Rewiring glucose metabolism improves 5-FU efficacy in p53-deficient/*KRAS^G12D* glycolytic colorectal tumors

Marlies C. Ludikhuize [1], Sira Gevers[1], Nguyen T. B. Nguyen[1], Maaike Meerlo[1], S. Khadijeh Shafiei Roudbari[1], M. Can Gulersonmez[1], Edwin C. A. Stigter[1], Jarno Drost [2,3], Hans Clevers [2,3,4], Boudewijn M. T. Burgering[1,4] & Maria J. Rodríguez Colman [1✉]

Despite the fact that 5-fluorouracil (5-FU) is the backbone for chemotherapy in colorectal cancer (CRC), the response rates in patients is limited to 50%. The mechanisms underlying 5-FU toxicity are debated, limiting the development of strategies to improve its efficacy. How fundamental aspects of cancer, such as driver mutations and phenotypic heterogeneity, relate to the 5-FU response remains obscure. This largely relies on the limited number of studies performed in pre-clinical models able to recapitulate the key features of CRC. Here, we analyzed the 5-FU response in patient-derived organoids that reproduce the different stages of CRC. We find that 5-FU induces pyrimidine imbalance, which leads to DNA damage and cell death in the actively proliferating cancer cells deficient in p53. Importantly, p53-deficiency leads to cell death due to impaired cell cycle arrest. Moreover, we find that targeting the Warburg effect in *KRAS^G12D* glycolytic tumor organoids enhances 5-FU toxicity by further altering the nucleotide pool and, importantly, without affecting non-transformed WT cells. Thus, p53 emerges as an important factor in determining the 5-FU response, and targeting cancer metabolism in combination with replication stress-inducing chemotherapies emerges as a promising strategy for CRC treatment.

[1] Molecular Cancer Research, Center for Molecular Medicine, University Medical Center Utrecht, 3584 CG Utrecht, the Netherlands. [2] Princess Máxima Center for Pediatric Oncology, 3584 CS Utrecht, The Netherlands. [3] Oncode Institute, Utrecht, The Netherlands. [4] Hubrecht Institute, Royal Netherlands Academy of Arts and Sciences, 3584 CT Utrecht, The Netherlands. ✉email: m.j.rodriguezcolman@umcutrecht.nl

Worldwide, colorectal cancer (CRC) is the third most commonly diagnosed type of cancer and the second leading cause of cancer-related mortality[1]. Surgery remains currently the only curative treatment in patients with early stage CRC or with resectable metastases. CRC patients additionally receive adjuvant chemotherapy and patients with unresectable, metastatic CRC entirely rely on chemotherapy[2]. Although 5-FU based chemotherapies have a poor tumor response (rates up to 50%)[3–5] and do not effectively extent the disease-free survival[2–4,6,7], it remains the most common treatment for CRC (reviewed in[8]). Furthermore, it remains unclear for which patients 5-FU therapy is beneficial.

Despite the importance of 5-FU, the underlying mechanism of its toxicity is still debated. Upon cellular uptake, 5-FU is converted into active fluorinated metabolites. 5-FUTP and 5-FdUTP can be incorporated into RNA and DNA respectively, and F-dUMP can inhibit thymidylate synthase (TS), impairing the deoxynucleotide pool and consequently DNA replication and repair (reviewed in ref.[8,9]). How this recapitulates in patients remains unclear, although a number of studies show that 5-FU can induce cytotoxicity via F-UTP incorporation into RNA[10–14]. TS expression in the tumor appears, on the other hand, to correlate with the 5-FU response, suggesting that 5-FU's toxicity could rely on impaired DNA replication and/or repair, however correlation does not imply causation[8,15–18].

Although precision medicine and application of targeted therapies have been facilitated by genomic studies, for conventional chemotherapies this has been relatively unsuccessful[2,19]. The relevance of different genetic mutations in determining 5-FU response, in fact, still remains elusive. Furthermore, genetic and phenotypic intra-tumor heterogeneity also contribute to differential therapy response and resistance[2,20–22]. Cancer stem cells (CSCs) are such a subset of tumor cells within the tumor that actively proliferate and exhibit differentiation potential[23–25]. Whether different CRC cell types respond differently to 5-FU also remains to be elucidated. Altogether, there is still a lack of knowledge that limits improvement of 5-FU-based CRC treatment strategies.

Studies aimed to increase our knowledge on the mechanisms of action of conventional chemotherapies should be performed in pre-clinical models that allow manipulation, but at the same time recapitulate the morphological and molecular characteristics of CRC tumors. Tumor-derived 2D cell lines have greatly contributed to the current understanding of cancer biology, but in most cases exhibit poor (genetic) stability and lack the cellular heterogeneity of tumors in vivo[26]. In contrast, tumor-biopsies that exhibit all features of the tumor, fall oftentimes short for investigation due to the limited material and lacking options for manipulation. Patient-derived organoids (PDOs) bridge the gap between patient-biopsies and 2D cell lines. PDOs recapitulate somatic copy number variations and mutation spectra found in CRC tumors and the genetic and non-genetic heterogeneity of CRC tumors[27,28]. Moreover, multiple studies have shown that tumor-derived organoids from multiple cancer types predict the chemotherapy response in patients[29–33] and the response is stable over time[32,34]. Furthermore, organoids allow manipulation to assess causality and the mechanisms downstream drug toxicity.

Tumors have a complex genetic background and not all genetic lesions are drivers of tumor progression and neither they determine therapy response[2,35,36]. To identify the specific mutations that determine chemotherapy response, we chose a well-defined system, the CRC tumor progression organoid model (TPO)[37] and PDOs. The TPO model consists of organoids derived from healthy colon tissue that have been genetically engineered to harbor the four most frequent driver mutations of CRC ($APC^{KO}$, $KRAS^{G12D}$, $P53^{KO}$, $SMAD4^{KO}$). PDOs are directly derived from

patient's tumors and have been previously characterized[27]. Here we show that p53-deficiency consistently provokes DNA damage and cell death upon 5-FU treatment, both in TPOs and PDOs. This occurs because of the inability of p53-deficient cells to halt cell proliferation. Active p53 protects against 5-FU-induced DNA damage through inducing G1 arrest in non-transformed WT and AK ($APC^{KO}$, $KRAS^{G12D}$) organoids leading to survival. In PDOs, we observe a more variable response; although p53 does induce cell cycle arrest and protects against 5-FU-induced DNA damage, it can additionally evoke a rapid apoptotic response. This differential response towards p53 is likely due to a complex interplay of p53 with the numerous additional genetic lesions present in PDOs. As we found that the 5-FU mode of action relies on pyrimidine imbalance, we targeted the metabolism of cancer cells to improve efficacy of the antimetabolite 5-FU. Of note, we found that rewiring the Warburg effect lowers the levels of nucleotides and enhances 5-FU toxicity selectively in p53-deficient and $KRAS^{G12D}$-glycolytic CRC cells but not in non-transformed intestinal cells.

## Results

**Inactive p53 determines 5-FU-induced DNA damage and cell death in human CRC organoids**. To link specific mutations to 5-FU sensitivity in CRC, we analyzed the 5-FU response in TPOs, which are genetically engineered to harbor different combinations of the main CRC drivers. We used in this study non-transformed (WT), $APC^{KO}KRAS^{G12D}$ (AK), $APC^{KO}P53^{KO}$ (AP), $APC^{KO}KRAS^{G12D}P53^{KO}$ (APK) and $APC^{KO}KRAS^{G12D}P53^{KO}SMAD4^{KO}$ (APKS)[37] TPOs. This model has been studied in vitro and in vivo and APK and APKS organoids recapitulate morphological features of respectively adenocarcinoma and poorly differentiated adenocarcinoma with metastatic potential[37,38]. First, we analyzed sensitivity to 5-FU treatment and found that the response was different across the different organoid lines. 5-FU significantly reduced cell viability in AP, APK, and APKS organoids, which also showed higher growth rates than WT and AK organoids. In contrast, WT and AK organoids showed no significant decrease in cell viability, although differences in organoid sizes were observed, suggesting a cytostatic effect of 5-FU (Fig. 1a, b and Supplementary Fig. 1a–d). P53 is a well-established factor and component of the DNA damage response and 5-FU can interfere with DNA synthesis[8,9,39]. Thus, we evaluated the DNA damage marker γH2AX upon 5-FU. Western blot and flow cytometry analysis showed that, 5-FU treatment induced DNA damage in the p53-deficient organoids, whereas this phenotype was milder and/or not significant in the WT and AK organoids (Fig. 1c, d and Supplementary Fig. 1d, e). In order to gain further insights into the 5-FU-induced DNA damage, we analyzed the activation of the different DNA damage response pathways. Stalled replication forks and single stranded DNA breaks are common consequences of conventional chemotherapies[40,41]. Hence, we first evaluated the ATR-Chk1 pathway, which is activated upon replication stress and single strand breaks[42]. Time course experiments revealed that 5-FU induces Chk1 activation in both WT and p53-deficient organoids as early as 24 h (Supplementary Fig. 1e). At 24 h, WT organoids showed a mild induction of DNA damage marker γH2AX. $P53^{WT}$ organoids cleared the damage within the next 24 h, while p53-deficient organoids showed accumulation of γH2AX at this later time point (Supplementary Fig. 1e). This suggests that while $P53^{WT}$ cells resolve 5-FU-induced DNA damage, p53-deficient organoids fail to do so. Interestingly, inhibition of ATR prevents Chk1 activation in both transformed and WT organoids, indicating that Chk1 activation is a result of replication stress (Fig. 1e)[43]. In line with that, inhibition of the ATR-Chk1 pathway enhances γH2AX levels in

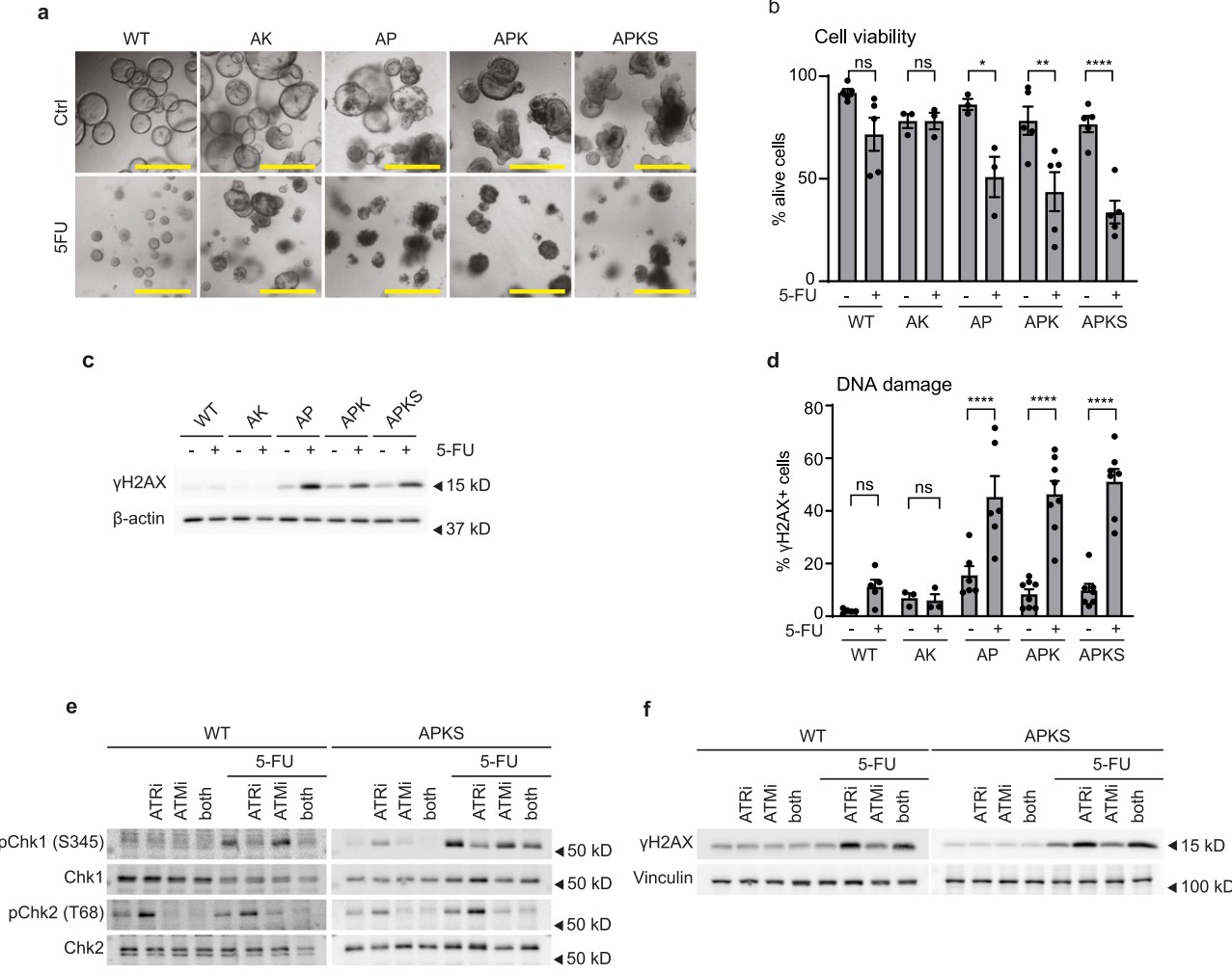

**Fig. 1 5-FU induces DNA damage and cell death in p53-deficient CRC organoids. a** Representative brightfield images of WT and CRC organoids treated with 5-FU for 7 days (scale bar = 500 µm). **b** Cell viability analysis of WT and CRC tumor organoids by flow cytometry to distinguish alive (DAPI⁻) from dead cells (DAPI⁺) upon 7 days of 5-FU treatment (mean ± SEM, $n = 3$–5, one-way ANOVA, Sidak's multiple comparisons test). **c** Western blot detection of γH2AX and β-actin in lysates from WT and CRC tumor organoids treated with 5-FU for 48 h (representative for $n = 5$). WT, AK, AP, APK, APKS were run together on the same gel. **d** Quantification of cells with DNA damage by flow cytometry of WT and CRC organoids treated with 5-FU for 48 h and stained with anti-γH2AX (mean ± SEM, $n = 5$ (WT), 3 (AK), 6 (AP), 8 (APK and APKS), one-way ANOVA, Sidak's multiple comparisons test). **e, f** Western blot detection of (p)Chk1, (p)Chk2, γH2AX and vinculin of lysates from WT and APKS organoids treated with 5-FU for 24 h and co-treated with either ATR inhibitor VE-821 or ATM inhibitor KU55933 or both for 26 h (representative for $n = 3$, WT: Chk2: Santa Cruz, #SC-9064, APKS: Chk2: Cell Signaling, #3440). WT and APKS were run on separate gels. ns: non-significant, $*p < 0.05$, $**p < 0.01$, $****p < 0.0001$.

both WT and AKPS organoids (Fig. 1f). During replication stress, unrepaired stalled replication forks can lead to double stranded DNA breaks and activation of the ATM-Chk2 pathway[42,44]. We found that 5-FU leads to ATM-dependent activation of Chk2 in APK and APKS, but not in WT, AK and AP organoids (Fig. 1e and Supplementary Fig. 1e)[45]. Taken together, these results show that ATR-Chk1 pathway is required for resolving 5-FU induced replication stress and importantly, that 5-FU treatment leads to increased levels of unresolved DNA damage and cell death in p53-deficient tumor organoids.

**5-FU induces DNA damage in proliferating cancer cells.** Our aforementioned results show that not all cells respond to the treatment uniformly, as a fraction of cells do not show DNA damage and survive treatment (Fig. 1b, d). To analyze the 5-FU response at a single cell level, we further investigated this on the 5-FU responsive organoids (AP, APK, and APKS). We first analyzed 5-FU-induced DNA damage by immunofluorescence and examined the response at the single cell level. Indeed, within

single organoids, 5-FU-induced DNA damage is heterogeneous between the cells (Fig. 2a and Supplementary Fig. 2a). Next to genetic heterogeneity, CRC tumors display phenotypic heterogeneity. Similarly to the healthy intestine, cancer stem cells (CSCs) are are marked by high Wnt signaling are proliferative and fuel tumor growth[23–25,46,47]. To examine the response to 5-FU in CSCs, we genetically introduced in our organoid lines the Wnt-based stem cell reporter STAR[48–51]. Both in WT and CRC organoids, cells showed heterogeneity in stemness (Fig. 2b and Supplementary Fig. 2b). Furthermore, EdU incorporation and cell cycle analyses showed that STAR⁺ cells in both in WT and CRC organoids, indeed constitute the proliferating population of cells (Supplementary Fig. 2c–e)[23,24]. Interestingly, flow cytometry and immunostaining analyses revealed that CSCs acquire more DNA damage than differentiated cells upon 5-FU treatment (Fig. 2c, d and Supplementary Fig. 2c, f), which relates to their high proliferation rate. In line with that, immunostaining of the proliferative marker Ki67 in combination with γH2AX, revealed that, upon 5-FU, Ki67⁺-proliferating cells have more DNA damage

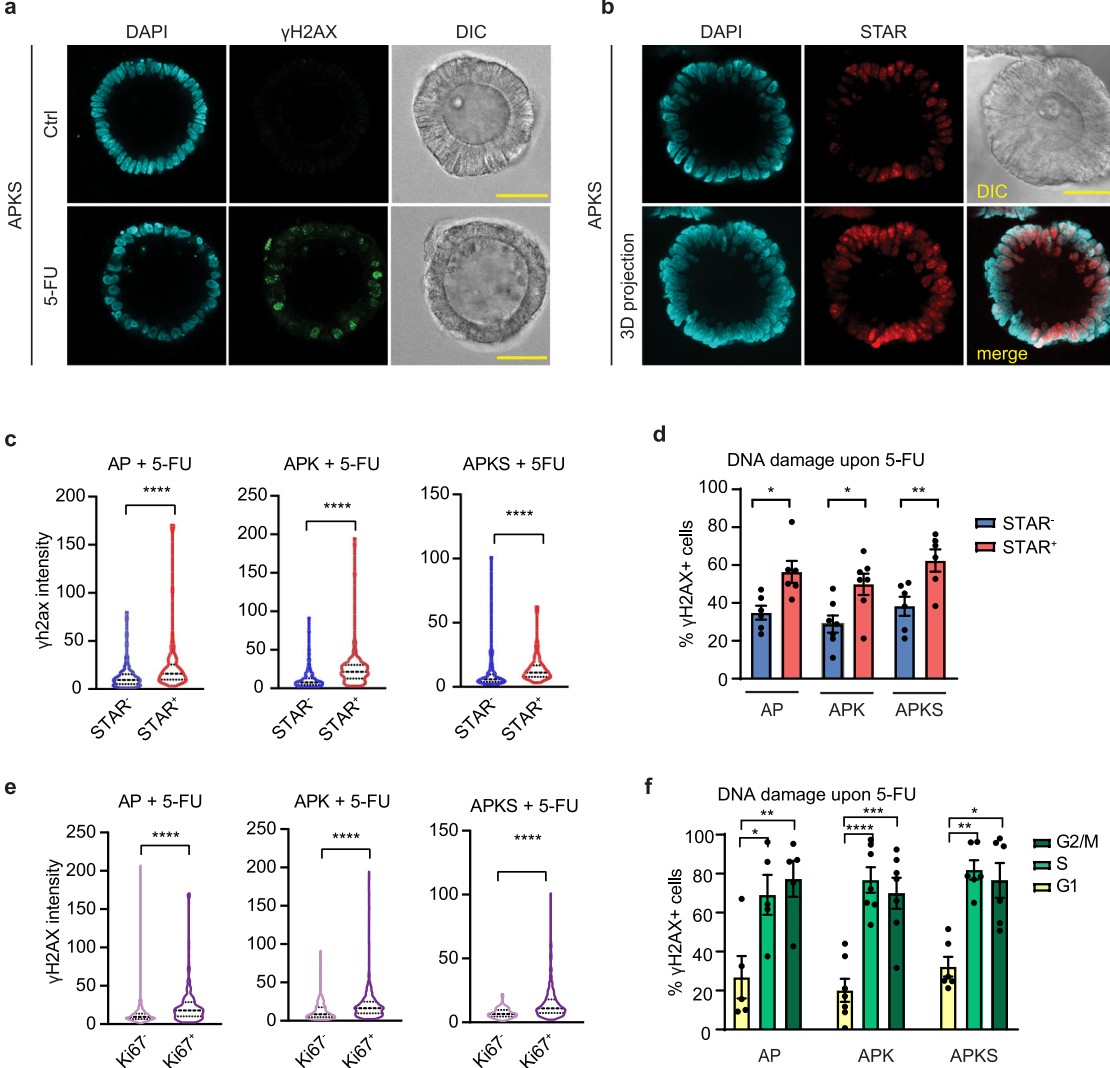

**Fig. 2 5-FU induces DNA damage in a proliferating subpopulation of tumor cells. a** Representative images of APKS organoids treated with 5-FU for 48 h and stained with anti-γH2AX and DAPI (scale bar = 50 μm). **b** Representative images of APKS organoids transduced with the stem cell reporter STAR and stained with DAPI (scale bar = 50 μm, upper panel: single Z-stack, lower panel: maximal projection of Z-stacks). **c** Quantification of γH2AX intensity in STAR⁻ and STAR⁺ cells of immunofluorescent images of AP, APK, and APKS organoids treated with 5-FU for 48 h (Supplementary Fig. 2f) (70-153 cells per condition from 12 organoids from three independent experiments, Mann–Whitney test). **d** Detection of γH2AX⁺ cells by flow cytometry in STAR⁻ vs STAR⁺ cells of AP, APK and APKS organoids treated with 5-FU for 48 h (mean ± SEM, n = 6 (AP, APKS), 7(APKS), one-way ANOVA, Sidak's multiple comparisons test). **e** Quantification of γH2AX intensity in KI67⁺ vs KI67⁻ cells of immunofluorescent images of AP, APK, and APKS organoids treated with 5-FU for 48 h (Supplementary Fig. 2G) 128-331 cells per condition from 12 organoids from three independent experiments, Mann–Whitney test).
**f** Detection of γH2AX⁺ cells by flow cytometry G1, S and G2/M cells of AP, APK and APKS organoids treated with 5-FU for 48 h (mean ± SEM, n = 5 (AP), 7 (APK, APKS), AP, APK: one-way ANOVA, Sidak's multiple comparisons test, APKS: Kruskal–Wallis test, Dunn's multiple comparisons test). ns: non-significant, *p < 0.05, **p < 0.01, ***p < 0.001, ****p < 0.0001.

than non-proliferating Ki67⁻ cells (Fig. 2e and Supplementary Fig. 2g). Furthermore, analysis of γH2AX in each of the cell cycle phases showed that the proportion of DNA damaged cells is higher in S and G2/M phase compared to cells in G1 (Fig. 2f). Interestingly, this pattern observed in cycling CSCs, is also found in the (small proportion) of cycling STAR⁻ differentiated cells (Supplementary Fig. 2h). These results indicate that an active proliferation state in the cells is critical for 5-FU sensitivity.

**Inactive p53 leads to 5-FU-induced DNA damage and cell death due to the lack of G1 arrest in TPOs and PDOs.** Although the importance of p53 in the 5-FU response has been investigated, a clear picture is yet to emerge[17,52–55]. While in vitro studies show that p53 is required for 5-FU-induced apoptosis[56–60],

epidemiological studies show that *P53* expression correlates with 5-FU resistance in stage III and IV patients[52,54,61]. Here, we find that p53 loss of function leads to 5-FU sensitivity and therefore we investigated the mechanism behind this phenotype. Western blot analysis showed that p53 and its transcriptional target p21 increase at respectively 4 and 16 h of 5-FU treatment in WT and AK organoids (Fig. 3a). P53 regulates cell proliferation via p21 that binds to and inhibits the cyclin/CDK complexes which phosphorylate Rb, thereby preventing its phosphorylation and consequent inhibition of G1/S transition (reviewed in[62]). We found that 5-FU treatment causes a complete loss of Rb phosphorylation in *P53^WT* organoids, whereas p53-deficient organoids lack of p21 induction and show remaining Rb phosphorylation (Fig. 3b). In agreement, cell cycle profile analysis showed that

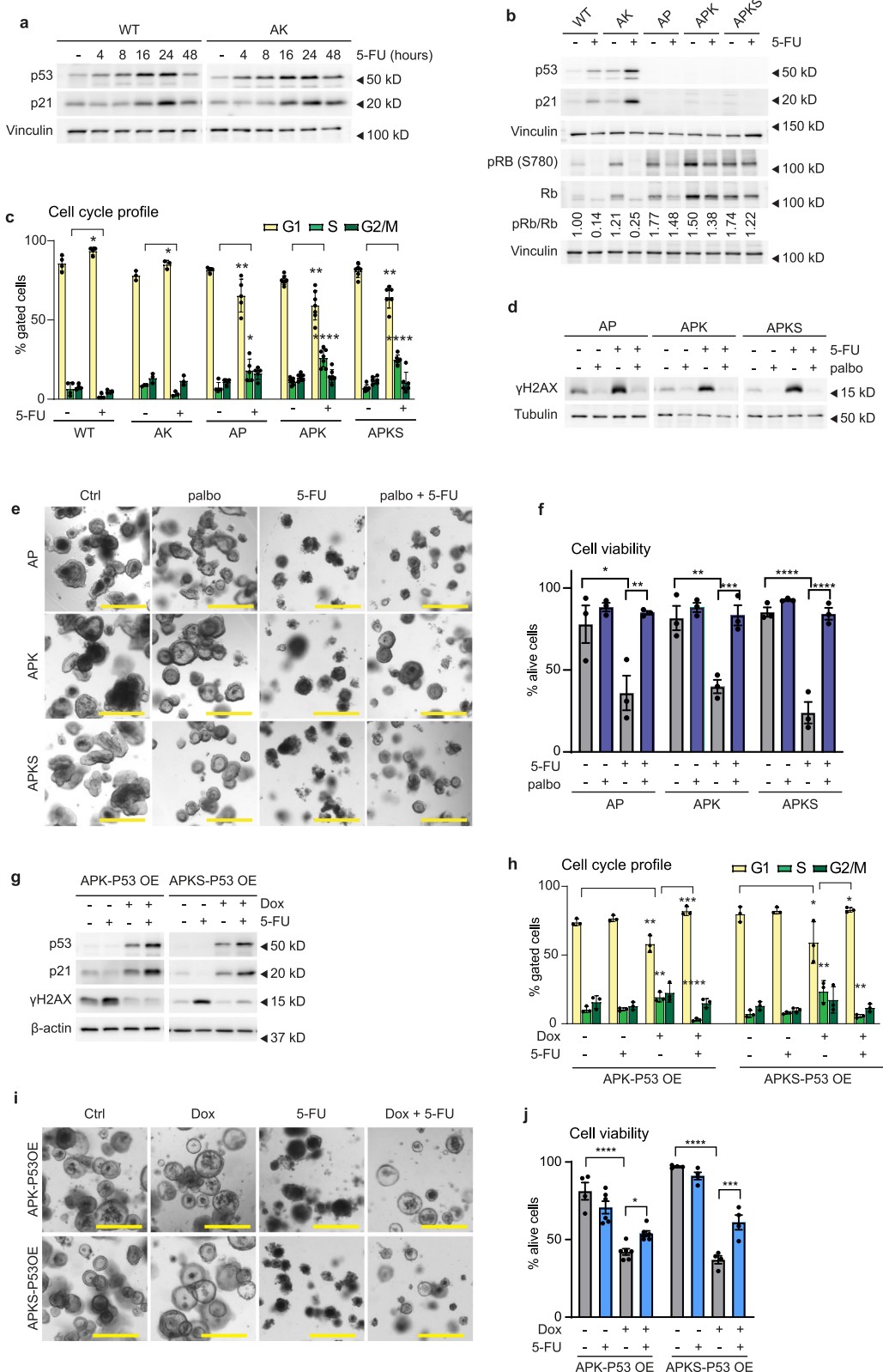

5-FU induces a G1 arrest in $P53^{WT}$ organoids, whereas it causes S/G2-phase accumulation in p53-deficient organoids (Fig. 3c, Supplementary Fig. 1d and Table 1). These points at p53-induced G1-arrest as the mechanism preventing 5-FU-induced DNA damage, which is in line with the observed reduced size in WT and AK (Fig. 1a and Supplementary Fig. 1b, c). To test that, we substituted p53 function by arresting the cells in G1 by

the CDK4/6 inhibitor palbociclib[63]. Upon 24 h of treatment, most cells were arrested in G1 (Supplementary Fig. 3a, b) and we proceeded with the 5-FU administration. G1 arrest in p53-deficient organoids completely prevented DNA damage upon 5-FU and rescued 5-FU-induced cell death (Fig. 3d–f), suggesting that DNA damage during replication is the main contributor to 5-FU toxicity in p53-deficient organoids. In order to further

**Fig. 3 Inactive p53 leads to 5-FU-induced DNA damage and cell death due to the lack of G1 arrest in TPOs. a** Western blot detection of p53, p21 and vinculin in WT and AK organoids treated with 5-FU for 4, 8, 16, 24, and 48 h (blot representative for $n = 4$). WT and AK were run on the same gel. **b** Western blot detection of p53, p21, (p)Rb and vinculin in WT and CRC organoids treated with 5-FU for 48 h (representative for $n = 4$). WT, AK, AP, APK, APKS were run together on the same gels. **c** Cell cycle profile determined by flow cytometry in WT and CRC organoids treated with 5-FU for 48 h (mean ± SEM, $n = 4$ (WT), 3 (AK), 5 (AP), 7 (APK), 6 (APKS), Kruskal–Wallis test, unpaired $t$-test and Mann–Whitney test). **d** Western blot analysis of γH2AX and tubulin of AP, APK and APKS organoids treated with 5-FU for 48 h. Palbociclib treatment was started 24 h before 5-FU treatment to arrest cells in G1 (representative for $n = 3$). AP, APK, and APKS were run together on the same gels. **e, f** Representative brightfield imaging (**e**) and cell viability analysis (**f**) of AP, APK, and APKS organoids treated with 5-FU for 6 days. Palbociclib treatment started 24 h before 5-FU treatment to arrest cells in G1 (scale bar = 500 μm, mean ± SEM, $n = 3$, one-way ANOVA, Sidak's multiple comparisons test). **g** Western blot detection of p53, p21, γH2AX, and β-actin in doxycycline-inducible APK-P53OE and APKS -P53OE organoids treated with 5-FU for 48 h (doxycycline (40 ng/ml for APK and 200 ng/ml for APKS) treatment was started 16 h before 5-FU administration) (representative for $n = 3$). APK and APKS were run on separate gels. **h** Cell cycle profiles of doxycycline-inducible APK-P53OE and APKS-P53OE organoids treated with 5-FU for 48 h (doxycycline (40 ng/ml for APK and 200 ng/ml for APKS) treatment was started 16 h before 5-FU administration) (mean ± SEM, $n = 3$, one-way ANOVA, Sidak's multiple comparisons test). **i, j** Brightfield images (**i**) and cell viability analysis (**j**) of doxycycline-inducible APK-P53OE and APKS-P53OE organoids by flow cytometry to distinguish alive (DAPI⁻) from dead (DAPI⁺) cells upon 4 days of 5-FU treatment. APK: doxycycline (40 ng/ml) treatment was started 16 h before and washed away 24 h after 5-FU administration. APKS: doxycycline (200 ng/ml) treatment was started 16 h before 5-FU administration and was not washed away (scale bar = 500 μm, mean ± SEM, $n = 6$ (APK-P53 OE), 4 (APKS-P53 OE), one-way ANOVA, Sidak's multiple comparisons test). ns: non-significant, $*p < 0.05$, $**p < 0.01$, $***p < 0.001$, $****p < 0.0001$.

**Table 1 Cell cycle phases analyzed by flow cytometry in WT and CRC organoids treated with 5-FU for 48 h.**

|  | G1 phase (%) | S phase (%) | G2/M phase (%) |
|---|---|---|---|
| WT CTRL | 85.8 (±4.8) | 6.5 (±4.1) | 7.8 (±1.7) |
| WT 5FU | 93.5 (±2.4) | 1.8 (±1.5) | 4.8 (±1.0) |
| AK CTRL | 78.0 (±3.0) | 8.7 (±0.6) | 13.3 (±3.1) |
| AK 5FU | 85.0 (±2.6) | 3.3 (±1.5) | 11.7 (±3.1) |
| AP CTRL | 81.6 (±1.5) | 7.4 (±3.1) | 11.0 (±1.7) |
| AP 5FU | 65.4 (±10.4) | 18.2 (±7.1) | 16.4 (±3.8) |
| APK CTRL | 75.1 (±2.8) | 11.1 (±2.1) | 13.7 (±2.1) |
| APK 5FU | 59.3 (±9.3) | 26.1 (±5.7) | 14.6 (±4.0) |
| APKS CTRL | 81.0 (±4.1) | 7.0 (±2.1) | 12 (±2.3) |
| APKS 5FU | 64.5 (±7.0) | 25 (±3.0) | 10.5 (±6.5) |

validate the function of p53 in the 5-FU response, we performed conditional *P53* add-back experiments using a doxycycline-inducible *P53* overexpression (OE) system (Fig. 3g–j and Supplementary Fig. 3c–f). Inducing *P53* reestablished the 5-FU response in APK and APKS, as p21 induction was restored and S/G2 cell cycle accumulation was prevented (Fig. 3g, h). In line with that, DNA damage and cell death were significantly rescued upon *P53* OE (Fig. 3g, i, j). Incomplete cell viability rescue could result from the add-back system, as it does not recapitulate the transient stabilization of p53 upon 5-FU (Fig. 3a) and sustained p53 activation can lead to apoptosis[62,64,65]. These results confirm that p53 acts as a discriminating factor in the 5-FU response, where loss of p53 causes 5-FU-induced DNA damage and cell death.

Next, we interrogated whether this mechanism is conserved in a model closer to the CRC patients. Thus, we assessed the 5-FU response in organoids derived from tumors of CRC patients (PDOs)[27]. We selected lines with or without mutations in *P53* and validated the p53 function based on the sensitivity to MDM2 inhibition (Nutlin-3)[27] (Supplementary Fig. 4a). Nutlin-3 induced cell death in P7t and P14t organoids, while P9t, P16t, P19bt were resistant, indicating functional and non-functional p53, respectively (Supplementary Fig. 4a). Upon 5-FU treatment, the p53-deficient lines P9t, P16t, P19bt responded similarly as the TPO model, as they were unable to induce p21 and G1 arrest and underwent DNA damage and cell death (Fig. 4a–e and Supplementary Fig. 1d). In PDO lines with functional p53 (P7t and P14t), we observed that 5-FU indeed induces p53 stabilization, p21 induction and G1 arrest (Supplementary

Fig. 4b, c). However, these PDOs rapidly underwent apoptotic cell death with no clear signs of DNA damage (Supplementary Fig. 4b, d, e). In contrast to TPOs, the PDOs harbor a considerable number of additional genetic lesions compared to the TPOs. This indicates that, in presence of these additional specific tumor-intrinsic conditions, such as high basal levels of p53 in P14t (Supplementary Fig. 4b) or the hypermutated state of P7t[27], p53 function can divert from mediating resistance to inducing apoptosis. To further study p53 function in PDOs, we introduced the p53 add-back system to p53-deficient lines (Supplementary Fig. 4f). Reintroduction of P53 in p19bt and p16t restored p21 induction, prevented S/G2 cell cycle accumulation and efficiently prevented DNA damage upon 5-FU treatment (Fig. 4f, g and Supplementary Fig. 4f). *P53* OE clearly rescued viability in P19bt and mildly in p16t organoids (Fig. 4h, i). Altogether, these results indicate that p53 loss of function in tumors has a consistent outcome characterized by DNA damage induced cell death. In *P53^WT^* tumors, 5-FU induces p53 activation, p21 induction and G1 arrest, and depending on the additional tumor intrinsic conditions, p53 can either protect or induce apoptosis in a DNA damage independent manner.

**5-FU induces DNA damage and cell death via a pyrimidine imbalance in p53-deficient organoids.** The mechanism of 5-FU-induced cytotoxicity is still debated (reviewed in refs. [8,9]). Here, we found that 5-FU-induced cell death relies on triggering DNA damage in cycling cells in organoids lacking functional p53, which indicates replication stress. Based on the reported effect of 5-FU on TS activity, we analyzed whether 5-FU-induced changes in the pyrimidine pool could explain this phenotype. First, we performed metabolomics analysis of APKS organoids, which showed the strongest response to 5-FU. We observed, that upon 5-FU, dUDP and dUMP levels increased, whereas the TDP and TTP pools were depleted (Fig. 5a), indicating 5-FU inhibits TS activity in CRC organoids. Cancer cells have recently been described to exhibit a nucleotide overflow mechanism that can balance out disrupted pyrimidine synthesis[66]. This nucleotide overflow mechanism involves deoxyuridine excretion to prevent dUMP accumulation and the rate of deoxyuridine excretion is dependent on the extent of TS inhibition[66]. Interestingly, we observed an increase in extracellular deoxyuridine levels upon 5-FU treatment (Fig. 5a), further supporting 5-FU induced TS inhibition. To examine the importance of the pyrimidine imbalance on DNA damage and cell death upon 5-FU, we performed nucleoside addback experiments. Administration of a mix of

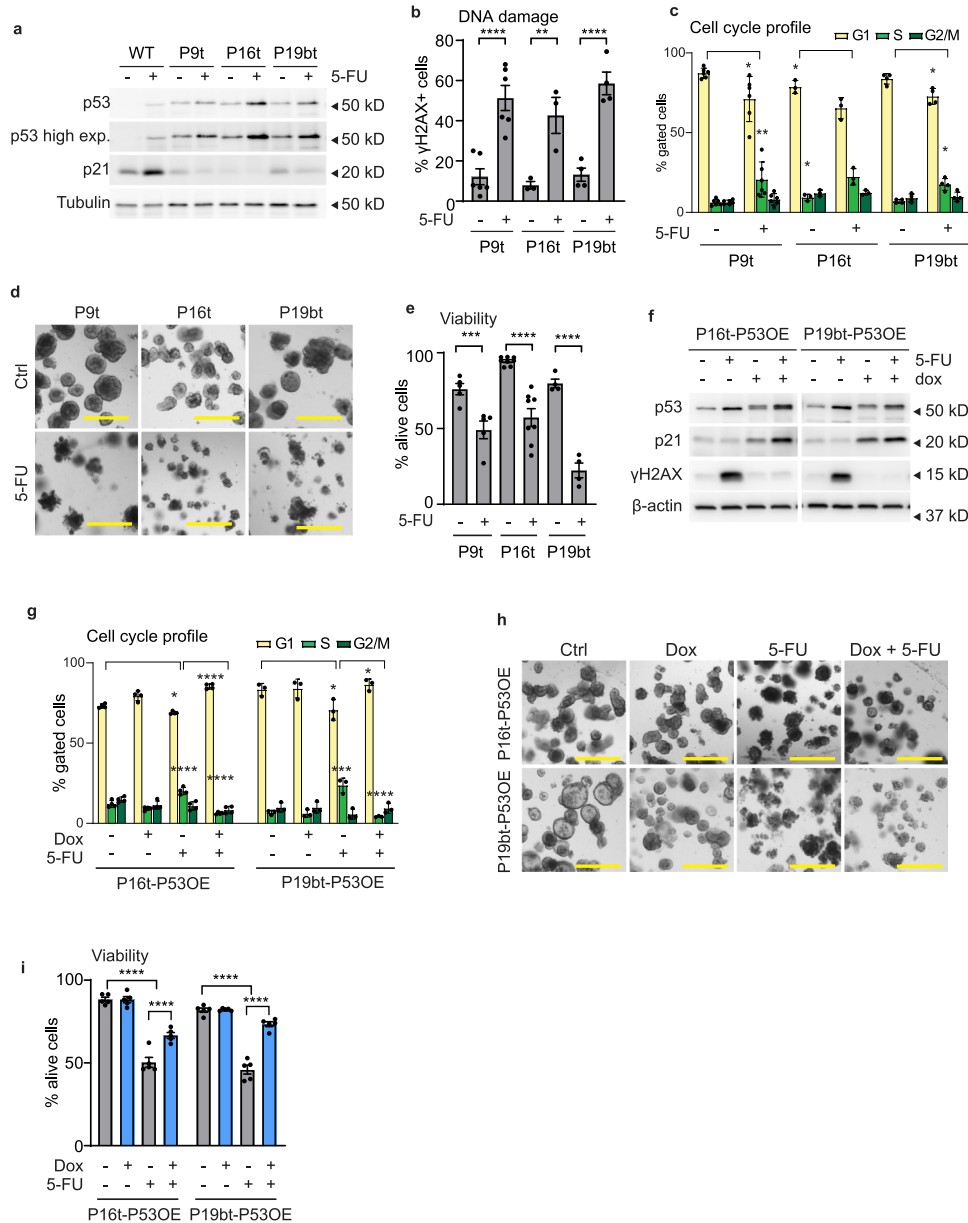

**Fig. 4 Inactive p53 leads to 5-FU-induced DNA damage and cell death due to the lack of G1 arrest in PDOs. a** Western blot detection of p53, p21 and tubulin in WT and P9t, P16t and P19bt organoids treated with 5-FU for 48 h (representative for $n = 3$). WT, P9t, P16t and P19bt were run together on the same gels. **b** Quantification of cells with DNA damage by flow cytometry of P9t, P16t and P19bt organoids treated with 5-FU for 48 h and stained with anti-γH2AX (mean ± SEM, $n = 6$ (P9t), 3 (P16t), 4 (P19bt), one-way ANOVA, Sidak's multiple comparisons test). **c** Cell cycle profile analysis by flow cytometry of P9t, P16t, and P19bt organoids treated with 5-FU for 48 h and stained with DAPI (mean ± SEM, $n = 6$ (P9t), 3 (P16t), 4 (P19bt) 4, P9t and P16t: unpaired $t$-test, P19bt: Mann–Whitney test). **d**, **e** Brightfield images (**d**) and cell viablility analysis (**e**) of P9t, P16t, and P19bt organoids treated with 5-FU for 6 days (scale bar = 500 µm, mean ± SEM, $n = 5$ (P9t), 8 (P16t), 4 (P19bt) 4, one-way ANOVA, Sidak's multiple comparisons test). **f**, **g** Western blot detection of p53, p21, γH2AX, and β-actin (**f**) and cell cycle profile analysis (**g**) in doxycycline-inducible P16t-P53OE and P19bt-P53OE organoids treated with 5-FU for 48 h. Doxycycline treatment (200 ng/ml) was started 16 h before and washed away 24 h after 5-FU administration (representative for $n = 3$, mean ± SEM, $n = 4$ (P16t-P53OE), 3 (P19bt-P53OE), one-way ANOVA, Sidak's multiple comparisons test). P16t-P53OE and P19bt-P53OE were run together on the same gels. **h**, **i** Bright field images (**h**) and cell viability analysis (**i**) of doxycycline-inducible P16t-P53OE and P19bt-P53OE organoids treated with 5-FU for 6 days. Doxycycline treatment (200 ng/ml) was started 16 h before and washed away 24 h after 5-FU administration (scale bar = 500 µm, mean ± SEM, $n = 5$, one-way ANOVA, Sidak's multiple comparisons test). *$p < 0.05$, **$p < 0.01$, ***$p < 0.001$, ****$p < 0.0001$.

nucleosides (adenosine, guanosine, cytidine, and thymidine) prevented 5-FU-induced DNA damage, S-phase accumulation and cell death in APKS organoids (Fig. 5b, c, e, f). Interestingly, single addition of thymidine, but not of the other nucleosides, was sufficient to rescue the 5-FU effect on cell cycle, DNA damage and cell death (Fig. 5b–f). Thymidine administration by itself did not affect cell cycle progression, showing that this rescue is not

dependent on cell cycle effects (Fig. 5d). Together these results indicate that, mechanistically, 5-FU induces DNA damage and cell death by altering the pyrimidine pool in p53-deficient organoids.

**Redirecting glucose metabolism lowers nucleotide levels and enhances the 5-FU-induced DNA damage.** Cancer cells undergo

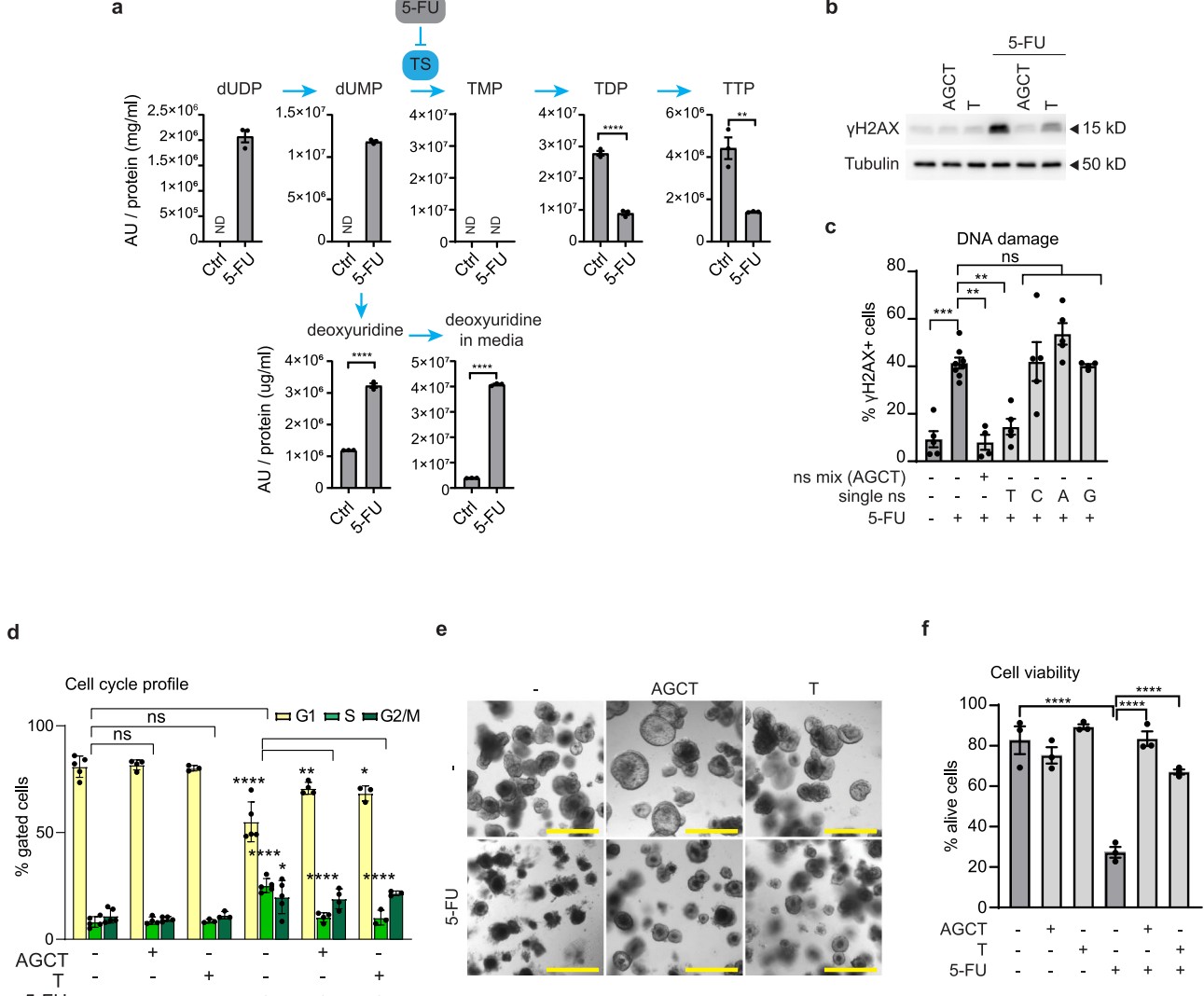

**Fig. 5 5-FU induces DNA damage and cell death by a pyrimidine imbalance. a** Detection of pyrimidines by metabolomics in APKS organoids treated with 5-FU for 30 h (mean ± SEM, *n* = 3 technical replicates, one-way ANOVA, Sidak's multiple comparisons test). **b** Western blot detection of γH2AX and tubulin of APKS organoids treated with 5-FU, a nucleoside mix (A, G, C, and T (25 μM each)), or T (25 μM) for 48hrs (blot representative for *n* = 3). All samples were run on the same gel. **c** Quantification of cells with DNA damage by flow cytometry of APKS organoids treated with 5-FU, a nucleoside mix (A, G, C and T (25 μM each)), or the separate nucleosides (25 μM) for 48 h (mean ± SEM, *n* = 5, 6 (5-FU), 4 (mix), 3 (G), one-way ANOVA, Sidak's multiple comparisons test). **d** Cell cycle profile analysis by flow cytometry of APKS organoids, treated with 5-FU, a nucleoside mix (A, G, C and T (25 μM each)), or T (25 μM) for 48 h (mean ± SEM, *n* = 5 (ctrl, 5-FU), 4 (+AGCT), 3 (+T), one-way ANOVA, Sidak's multiple comparison's test).
**e**, **f** Representative Brightfield images (**e**) and cell viability analysis (**f**) of APKS organoids treated with 5-FU, a nucleoside mix (adenosine, guanosine, cytosine and thymidine (25 μM each)), or thymidine (25 μM) for 6 days (scale bar = 500 μm, mean ± SEM, *n* = 3, one-way ANOVA, Sidak's multiple comparisons test). A adenosine, AU arbitrary unit, C cytidine, dUDP deoxyuridine diphospate, dUMP deoxyuridine monophosphate, G guanosine, ND not detected, ns mix nucleoside mix, ns non-significant, T thymidine, TMP thymidine monophosphate, TDP thymidine diphosphate, TTP thymidine triphosphate, TS thymidylate synthase. ns: non-significant, *p < 0.05, **p < 0.01, ***p < 0.001, ****p < 0.0001.

the Warburg effect by avidly taking up of glucose and metabolize it through glycolysis into lactate independently of oxygen availability. This increased glycolysis enables rapid production of ATP and supports the activity of anabolic pathways that rely on glycolytic intermediates, such as nucleotide synthesis. Based on the importance of nucleotides in the 5-FU toxicity, we rationalized that targeting the Warburg effect could improve 5-FU efficacy. Seahorse bioenergetics analysis showed that AK, APK, and APKS organoids have higher glycolytic rates than WT and AP organoids[67] (Fig. 6b), whereas the respiratory parameters are not significantly different (Supplementary Fig. 5a, b). These results show that the Warburg effect is recapitulated in in vitro organoids. Of note, these results point at constitutive active Ras

signaling (*KRAS*^G12D), rather than the p53 loss of function, as the main driver of the Warburg effect in CRC.

In order to lower the high glycolytic rates in tumor organoids, we first administered 2-deoxyglucose (2-DG), a glucose analogue that cannot be metabolized and accumulates in the cell leading to reduced glycolysis by the inhibition of hexokinase-2 (HK2; Fig. 6a; reviewed in ref. [68]). Although 2-DG efficiently decreased glycolysis (Supplementary Fig. 5c), in combination with 5-FU, it did not increase 5-FU-induced DNA damage and even lowered the effectiveness compared to 5-FU only treatment (Fig. 6c). EdU incorporation analysis revealed that 2-DG causes a major drop in proliferation (Fig. 6d), suggesting that targeting glycolysis while reducing proliferation does not enhance the 5-FU effect. Next, we

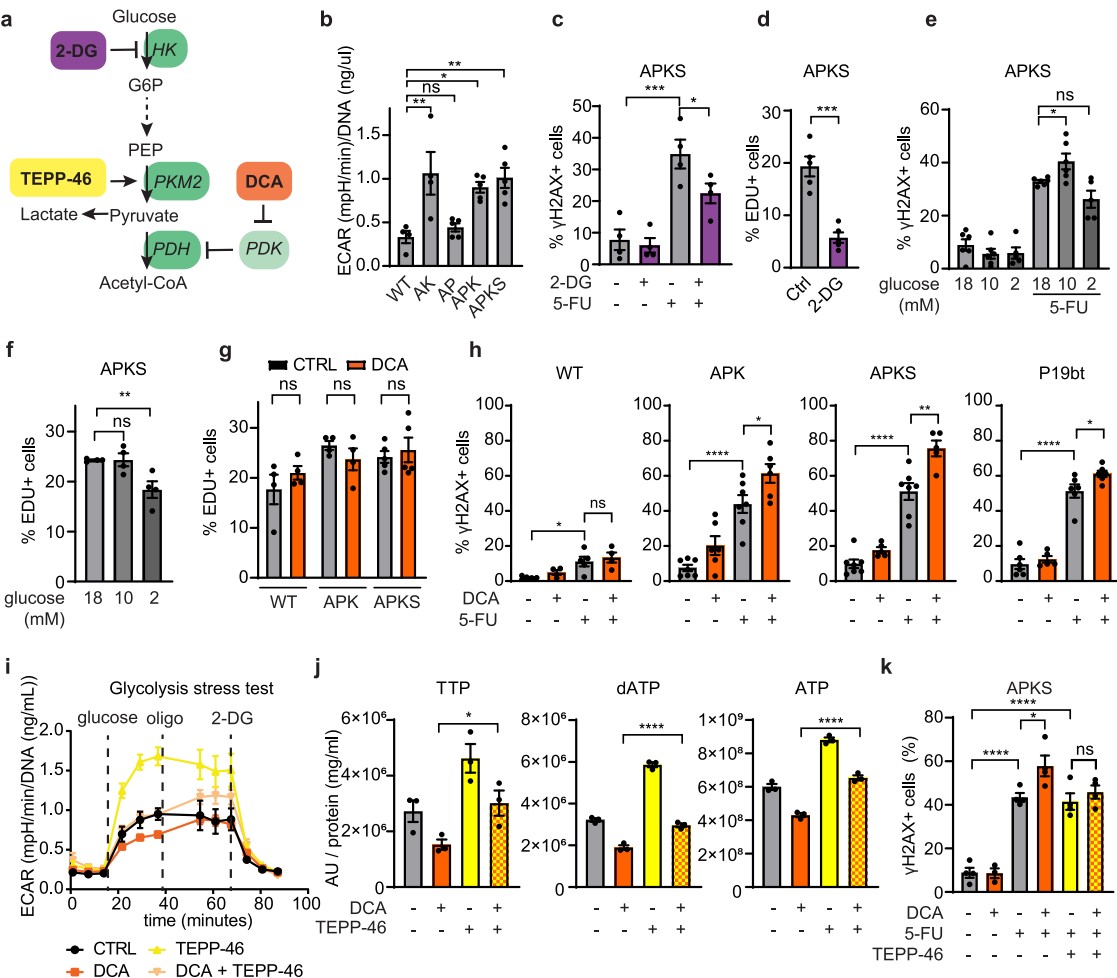

**Fig. 6 Redirecting glucose metabolism increases 5-FU-induced DNA damage. a** Schematic representation of glycolysis and different glycolysis-targeting drugs. **b** Extracellular acidification rate (ECAR) of WT and CRC organoids determined by a glycolysis stress test by Seahorse XF analysis (mean ± SEM, $n = 4$ (WT, AK), 5 (AP, APK, APKS), one-way ANOVA, Sidak's multiple comparisons test). **c** Quantification of cells with DNA damage by flow cytometry of APKS organoids treated with 5-FU for 48 h. 2-DG treatment started 20 h before 5-FU treatment (mean ± SEM, $n = 4$, one-way ANOVA, Sidak's multiple comparisons test). **d** EdU incorporation analysis by flow cytometry of APKS organoids treated with 2-DG for 24 h (mean ± SEM, $n = 5$, unpaired $t$-test). **e** Quantification of cells with DNA damage by flow cytometry of APKS organoids treated with 5-FU for 48 h. Glucose starvation started 20 h before 5-FU treatment (mean ± SEM, $n = 6$, 5 (2 mM glucose), one-way ANOVA, Sidak's multiple comparisons test). **f** EdU incorporation analysis by flow cytometry of APKS organoids starved for glucose for 24 h (mean ± SEM, $n = 4$, one-way ANOVA, Sidak's multiple comparisons test). **g** EdU incorporation analysis by flow cytometry of WT, APK, and APKS organoids treated with DCA for 24 h (mean ± SEM, $n = 4$, 5 (APKS), one-way ANOVA, Sidak's multiple comparisons test). **h** Quantification of cells with DNA damage by flow cytometry of WT, APK, APKS, and P19bt organoids treated with 5-FU for 48 h. DCA treatment (20 mM for TPOs and 10 mM for P19bt) started 20 h before 5-FU treatment (mean ± SEM, $n = 5$, 4 (WT), 6 (AK), 7 (APKS -/- and 5-FU), one-way ANOVA, Sidak's multiple comparisons test). **i** Determination of extracellular acidification rate (ECAR) during a Seahorse XF glycolysis stress test of APKS organoids treated with DCA and TEPP-46 for 24 h (mean ± SEM, five technical replicates, representative for $n = 3$). **j** Detection of TTP, dATP, and ATP by metabolomics of APKS organoids treated with DCA and TEPP-46 for 24 h (mean ± SEM, three technical replicates (from repeated measurements), one-way ANOVA, Sidak's multiple comparisons test). **k** Quantification of cells with DNA damage by flow cytometry of APKS organoids treated with 5-FU for 48 h. DCA and TEPP-46 treatments started 20 h before 5-FU treatment (mean ± SEM, $n = 4$, 3 (DCA), one-way ANOVA, Sidak's multiple comparisons test). 2-DG 2-deoxyglucose, ATP adenosine triphosphate, dATP deoxyadenosine triphosphate, DCA dichloroacetate, G6P glucose 6-phosphate, HK Hexokinase, PDH pyruvate dehydrogenase, PDK pyruvate dehydrogenase kinase, PEP phosphoenol pyruvate, PKM2 pyruvate kinase M2, TTP thymidine triphosphate. ns: non-significant, *$p < 0.05$, **$p < 0.01$, ***$p < 0.001$, ****$p < 0.0001$.

lowered glucose availability. Indeed, a limited reduction in glucose availability prior to 5-FU treatment increased DNA damage (Fig. 6e). However, further lowering glucose concentration reduced cell proliferation and, consequently, 5-FU efficacy (Fig. 6e, f). Thus, we looked for pharmacological options to reduce glycolysis in tumor organoids. DCA is an inhibitor of pyruvate dehydrogenase kinase (PDK), which increases pyruvate dehydrogenase (PDH) activity and consequently the conversion of pyruvate into acetyl-coA at the expense of lactate production (Fig. 6a). DCA treatment decreased PDH phosphorylation in all organoid lines

(Supplementary Fig. 5d), confirming that inhibits PDK activity. DCA reduced the high glycolytic rates of organoids that display the Warburg effect to glycolytic rates that are comparable to those of WT organoids, while increasing respiratory parameters (Supplementary Fig. 5e–g). Importantly, we did not find significant changes in proliferation upon DCA treatment for 24 h (Fig. 6g). Next, we assessed whether DCA altered nucleotide metabolism by metabolomics and found that DCA treatment indeed lowered nucleotide levels (Fig. 6j and Supplementary Fig. 5h). Importantly, administration of DCA as a short pretreatment to 5-FU indeed

increased the number of cells that undergo DNA damage upon 5-FU treatment in glycolytic APK and APKS organoids, whereas this could not be observed in the less glycolytic AP organoids (Fig. 6h and Supplementary Fig. 5i). Interestingly, similar results were obtained in our PDO models. DCA inhibited glycolysis in P19bt showing the Warburg effect and enhanced 5-FU-induced DNA damage (Fig. 6h and Supplementary Fig. 5j–l). In line with AP, in p16t no additional effects were observed by DCA treatment, as these lines do not show the Warburg effect phenotype (Supplementary Fig. 5j, k, m).

Next, we evaluated whether the synergistic effect of DCA on 5-FU treatment indeed relies on the inhibition of glycolysis. Proliferating cancer cells exhibiting the Warburg effect often express the pyruvate kinase M2 isoform (PKM2)[69]. As PKM catalyzes the rate-limiting step of glycolysis, we evaluated whether TEPP-46, a specific activator of PKM2 could rescue the effects of DCA treatment (Fig. 6a)[70]. To that end, we analyzed the glycolytic rates by Seahorse and the intracellular levels of glucose by live imaging of a glucose FRET sensor[71]. These analyses showed that PKM2 activation reverts glycolysis inhibition and restores intracellular glucose levels upon DCA treatment (Fig. 6i and Supplementary Fig. 5n–p). Importantly, in agreement with these results, TEPP-46 treatment rescued the decrease in nucleotides resulting from DCA treatment and indeed prevented the additive effect of DCA on 5-FU-induced DNA damage (Fig. 6j, k). Altogether, this shows that DCA enhances the 5-FU-induced DNA damage through lowering nucleotide levels as a consequence of the inhibition of the Warburg effect.

**Redirecting glucose metabolism enhances 5-FU-induced cell death.** Next, we evaluated whether DCA in combination with 5-FU can further increase cell death. To that end, we treated WT and 5-FU responsive organoids (APK, APKS, and P19bt) with 5-FU and 5-FU/DCA. Qualitative analysis showed that while upon 5-FU treatment some organoids survive the treatment, those were absent in the DCA/5-FU treatment (Fig. 7a and Supplementary Fig. 6a). We quantitatively analyzed this phenotype in bulk by flow cytometry and at the single organoid level, by calculating a viability score based on imaging analysis. In both cases, we found that DCA in combination with 5-FU leads to reduced cell viability when compared to single 5-FU treatment in APK, APKS and P19bt organoids and this additive effect of DCA was absent in WT organoids (Fig. 7a–c and Supplementary Fig. 6b, c). As a proxy for post-treatment survival and relapse potential, we evaluated the outgrowth capacity posterior to treatment. We treated organoids with 5-FU or with DCA/5-FU, removed treatment to allow organoid recovery and replated them to analyze organoid formation capacity. We found that the DCA/5-FU combination leads to a fewer number of formed organoids when compared to the 5-FU treatment in both APK and APKS organoids (Fig. 7d, e and Supplementary Fig. 6d). Thus, targeting the Warburg effect, by rewiring glucose metabolism, in combination with 5-FU increases DNA damage and cell death in highly glycolytic p53-deficient tumors without affecting non-transformed cells.

## Discussion

Here we use two models of human-derived organoids to dissect the mode of action of 5-FU and to understand the relevance of the different driver mutations on determining 5-FU efficacy in colorectal tumors. We found that when tumors harbor non-functional p53, they become sensitive to 5-FU through the induction of DNA damage and consequent cell death. Interestingly, in this model, the efficacy of 5-FU is dependent on the active proliferation state of cells and, mechanistically, 5-FU acts through inhibiting TTP synthesis. To enhance 5-FU efficacy in

p53-deficient and glycolytic tumors, we targeted the Warburg effect by redirecting glucose metabolism with the purpose of further altering the nucleotide pool. This strategy indeed improved 5-FU efficacy in those already sensitive p53-deficient and glycolytic tumors and importantly, it showed no additional toxicity to healthy non-transformed cells.

Precision medicine aims at patient stratification for treatment based on genetics, in a way that cancer patients receive the most adequate treatment[72]. However, for conventional chemotherapies, such as 5-FU, this has been unsuccessful, partially due to the unresolved understanding of the mechanisms of action[2,19]. Accumulating evidence points at 5-FU inducing cytotoxicity via both DNA and RNA[8–14,73]. This apparent differential mode of action could arise from the dose of 5-FU (ranging from 1 to 1000 μM in these studies), where high 5-FU levels have been proposed to target cells through RNA toxicity, whereas prolonged exposure to low doses is proposed to be cytotoxic via TS inhibition-induced DNA damage[8,74–76]. In patients the (low) 5-FU concentrations in plasma and tumors (8 μM and 45 μM/kg respectively[77]), suggest that the DNA damaging effect more likely causes toxicity in CRC tumors. In line with that, TS expression and the 5-FU response do correlate, where tumors with high TS levels are commonly more resistant to 5-FU therapy than tumors with low TS levels[8,15–18]. In the CRC organoids, we find that upon p53 deficiency 5-FU induces cytotoxicity in cycling cells mainly via impaired pyrimidine synthesis-induced DNA damage accumulation. Although we have not observed 5-FU-induced cytotoxicity in non-cycling cells, different 5-FU concentrations or timings could reveal alternative mechanisms of 5-FU-induced cytotoxicity.

Identifying a role for specific driver mutations in drug efficacy is challenging[2,35,36]. Here, we find that deficient p53 activity consistently ensures 5-FU toxicity through DNA damage-induced cell death. On the other hand, when p53 is functional, p53 is stabilized and induces p21, which results in cell cycle arrest that prevents 5-FU-induced DNA damage. We find that this response is either sufficient to protect against 5-FU and leads to survival, which is observed in the non-transformed WT and in AK organoids, or induces rapid apoptotic cell death with no signs of DNA damage, such as in the P7t and P14t organoids. Previous studies have shown two scenarios for p53: a cell cycle arrest-dependent protective role against 5-FU[56,78,79], but also that p53 is required to induce 5-FU-dependent apoptosis[14,56–60]. Our observations indicate that p53-induced apoptosis is likely independent of DNA damage and hence could be caused by other 5-FU-induced stresses such as RNA toxicity[8–14,73]. The balance between p53-induced cell cycle arrest and apoptosis could be dependent on 5-FU dose, or tumor intrinsic factors (epigenetic state, active signaling pathways) and cellular context (cellular microenvironment)[80–83], or in line with our results, on the presence of additional mutations. For instance, a recent study shows that oncogenic HRAS can lower the p53-dependent transcriptional response to replication stress[84].

CSCs were defined as a subpopulation of cells within a tumor that have high tumor initiating capacity. Therefore, originally it was suggested that CSCs are responsible for resistance to therapy and tumor relapse[21]. However, the occurrence of cellular plasticity by which non-CSCs can gain a CSC phenotype[24,46], indicates that therapy resistance is unlikely to be attributed to a specific cell type. In CRC tumors, most CSCs show high Wnt signaling and active proliferation[23–25,46]. Here, we find that cycling cells are efficiently targeted by 5-FU. Interestingly, we observed a population of stem cells in G1 that do not accumulate DNA damage. This could be in line with the previously reported subpopulation of slow cycling/quiescent CSCs within the CSC subpopulation showing increased resistance and relapse potential in CRC

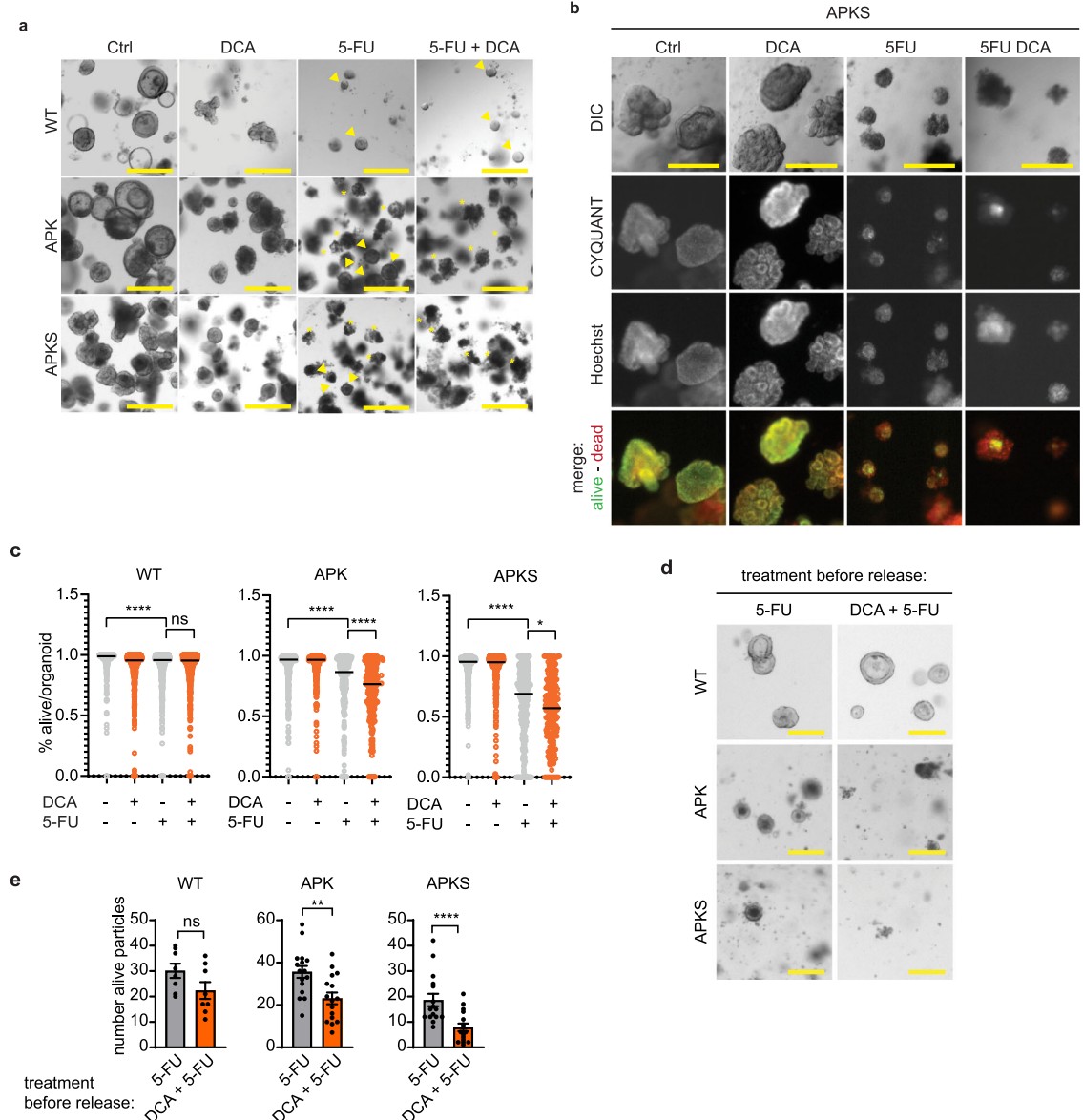

**Fig. 7 Redirecting glucose metabolism improves 5-FU-induced cytotoxicity. a** Representative bright field images of WT, APK, and APKS organoids treated with 5-FU for 7 days. DCA treatment was started 20 h before 5-FU administration (scale bar = 500 μm, arrow heads indicate survivor organoids, asterisks indicate compromised organoids). **b** Representative images of APKS organoids, stained with CYQUANT (alive) and Hoechst (total), treated with 5-FU for 7 days. DCA treatment started 20 h before 5-FU treatment (scale bar = 300 μm). **c** Quantification of alive score of images from b and Supplementary Fig. 6c (median, 237–720 organoids from three independent experiments, Kruskal–Wallis test, Dunn's multiple comparisons test). **d** Representative bright field images of WT, APK and APKS organoids 4 days after replating, upon a 48 h-5-FU treatment (50 μM) (with or without DCA treatment that started 20 h before 5-FU administration), followed by 7 days of recovery time (scale bar = 200 μm). **e** Determination of alive organoid particles based on Supplementary Fig. 6d (mean ± SEM, WT: 8 Matrigel droplets from two independent experiments, APK and APKS: 16 Matrigel droplets from 4 independent experiments, WT and APK: unpaired t-test, APKS: Mann–Whitney test). ns: non-significant, *$p < 0.05$, **$p < 0.01$, ****$p < 0.0001$.

tumors[85]. Together, these results stress that cellular behavior rather than a specific cell type determines 5-FU-sensitivity[81].

The metabolism of cancer cells is altered to support uncontrolled proliferation (reviewed in refs. [86–89]). The role of the Warburg metabolism (high rate of glycolysis towards lactate in the presence of sufficient oxygen) has been recently revised and it is proposed that lactate as the end product balances the cellular redox state to favor anabolic pathways (reviewed in refs. [86,87]). DCA is an FDA-approved drug (with minimal side effects) that is currently used for the treatment of metabolic diseases[90,91]. A previous study proposes that DCA could restore chemotherapy sensitivity to cells with acquired resistance through an elusively defined metabolic mechanism[92,93]. Here, we show that when applied as a short pretreatment to 5-FU, it

changes glucose metabolism and improves efficacy. DCA can only induce this additive effect in organoids that show both loss of functional p53 and exhibit the Warburg effect. Our results and previous evidence[94] suggest that $KRAS^{G12D}$ mutation, rather than p53 loss of function, results in the metabolic changes referred to as the Warburg effect in CRC. This point towards a subgroup of tumors that could benefit from this combination of treatment. Mechanistically, we show that DCA improves 5-FU efficacy by lowering the nucleotide levels as a consequence of inhibiting the Warburg effect, without inducing cytostasis. This is an additional mechanism to the reduced growth rates previously observed upon prolonged DCA treatment[92], that could be due to differences in proliferation and differentiation processes downstream metabolic transitions[95,96].

Based on this action of DCA, we propose that rewiring glucose metabolism likely also improve the efficacy of other chemotherapies that targets DNA replication.

While precision medicine moves forward, conventional chemotherapies are still the workhorse of oncology. A better understanding of their mode of action is key to find 'low cost-low toxicity' strategies to improve patient treatment. Considering that rewiring glucose metabolism does not appear to have adverse effects in healthy tissue, it emerges as a promising strategy of improving conventional chemotherapies.

## Methods

**Organoid culture.** Tumor progression organoids were a gift from the Clevers lab[37]. Patient-derived organoids P7t, P9t, P14bt, P16t and, P19bt were obtained from the HUB biobank and were characterized previously[27]. Organoids were cultured at 37 °C and at 5% $CO_2$. A mycoplasma-free status was confirmed routinely. The basic culture medium contained advanced DMEM/F12 supplemented with penicillin/streptomycin, 10 mM HEPES and 20 mM Glutamax. For experiments, upon trypsinization into single cells/small clumps of cells, cells were cultured in Matrigel (Corning, #356231) or BME (Bio-Techne, #3533-010-02) in expansion medium (Table 2), supplemented with Rock inhibitor Y-27632 (Gentaur, #607-A3008).

*Experimental set-up.* After 7 days, organoids were diluted and replated in 24-well plates (4 droplets of 11 µl) and medium was replaced to differentiation medium (Table 3). After 3 days, differentiation medium was refreshed and after another 20 h, 5-FU (Sigma, #F6627) was added. For all experiments 100 µM of 5-FU was used, unless stated differently. The timing of the 5-FU treatments is stated in the figure legends.

Treatments with ATR inhibitor VE-821 (5 µM, Bioconnect, #S8007) and ATM inhibitor KU-55933 (10 µM, Sigma, #SML1109) were started 2 h before 5-FU administration. Nucleosides (25 µM each, all Sigma: adenosine #A4036, thymidine #T9250, guanine #G6264 and cytidine #C4654) were added together with 5-FU. Treatments with palbociclib (1 µM, selleckchem, #S1116)) were started 24 h before 5-FU addition. DCA (20 mM, Sigma, #347795), 2-DG (10 mM, Sigma #D8375) and TEPP-46 (100 µM, Selleckchem, #S7302 treatments) or glucose starvations were started 20 h before 5-FU treatment. Glucose starvation medium was prepared with SILAC Advanced DMEM/F-12 Flex Media (Gibco, #A2494301), supplemented with penicillin/streptomycin, 10 mM HEPES and 20 mM Glutamax, L-Arginine (147.5 mg/L, Sigma, #A6969), L-Lysine (91.25 mg/L, Sigma, L8662) and the stated glucose (Merck Millipore, #1.08337.1000) concentration. For EdU incorporation analysis, 6 h prior to collection organoids were incubated with 1 µM EdU (Thermo fisher, #C10636). Details for doxycycline (Sigma, #D9891) and Nutlin-3 (Sanbio, #10004372) concentrations and incubation time are stated in the figure legends.

**Lentiviral transduction.** The pInducer-mKate2-NLS-P2A-P53 was a gift from the Snippert lab. From the Stem cell ASCL2 reporter (STAR) plasmid[48,50], the 8x STAR-sTomato-NLS sequence was cloned into a lentiviral vector with a puromycin-resistance cassette. The pcDNA3.1 FLII12Pglu-700uDelta6 (Addgene plasmid #17866) was a gift from Wolf Frommer[71]. The eCFP sequence was replaced for a codon optimized eCFP sequence to prevent recombination with the YFP sequence. The new glucose sensor sequence was cloned into a lentiviral vector under the control of a Hef1 promoter and with a puromycin resistance cassette. These constructs, together with third generation packaging vectors, HEK293T cells and LentiX Concentrator (Clontech) were used to generate and concentrate lentiviral particles. Organoids were lentivirally transduced as described in ref. [97]). In brief, organoids were trypsinized and incubated with concentrated virus (60 min while centrifuging at 600 rpm at RT followed by 4 h at 37 °C). Next, organoids were plated in Matrigel.

**Protein lysates and western blot.** Organoids were washed once with and collected in ice-cold PBS, supplemented with 5 mM NaF (Vwr, #1.06449.0350) and 1 mM $NaVO_3$ (Sigma, #S6508). Organoids were centrifuged and pellets were incubated in Cell Recovery Solution, supplemented with NaF and $NaVO_3$ on ice for 15 minutes. Upon centrifugation, pellets were lysed in lysis buffer (50 mM Tris pH 7.0, 1%TX-100, 15 mM $MgCl_2$, 5 µM EDTA, 0.1 mM NaCl, 5 mM NaF, 1 mM $NaVO_3$, 1 µg/mL Leupeptin (Sigma, #11034626001) and 1 µg/mL Aprotinin (Sigma, #10981532001) and protein content was determined by Biorad protein assay (#500-0006). Samples were adjusted for the protein content and Laemli sample buffer was added. Proteins were run in SDS-PAGE and transferred to Immobilon Polyscreen PVDF transfer membranes (#IPVH00010, Merck Millipore) or Amersham Protan nitrocellulose membranes (#10600001 GE Healthcare Life Sciences). Western blot analysis was performed with primary antibodies recognizing: vinculin (1:10,000, Sigma, #V9131), Tubulin (1:5000, Merck Millipore, #CP06 OS), γH2AX (1:10,000, Sigma, #05-636), pChk1(S345) (1:2000, Cell Signaling, #2348, Chk1 (1:1000, Santa Cruz, #SC-8408), pChk2(T68) (1:1000, Cell Signaling, #2661), Chk2 (1:1000, Cell Signaling, #3440 and Santa Cruz, #SC-9064), pRb(S780) (1:2000, Cell Signaling, #9307), Rb (1:1000, Santa Cruz, #SC-7905), p53 (1:1000, Santa Cruz, #SC-126), p21 (1:1000, BD Bioscience, #556430), pPDH(S293) (1:1000, Abcam, #ab92696), and PDH (1:1000, Invitrogen, #459400). Secondary HRP-conjugated antibodies targeting mouse and rabbit IgG were purchased from Biorad (1:10,000).

**Flow cytometry.** Organoids were collected in ice-cold DMEM/F12 medium, supplemented with penicillin/streptomycin, 10 mM HEPES and 1x Glutamax (DMEM/F12 +++ medium) and subsequently incubated with trypsin (Sigma). For cell viability analysis, cells were stained with DAPI (Sigma-Aldrich, #D9564) on ice for 5 min and immediately analyzed by flow cytometry. Viability was determined by DAPI staining, where $DAPI^-$ cells were considered alive, and $DAPI^+$ cells as dead.

For γH2AX staining, cell cycle profile analysis and EdU incorporation analysis, single cells were fixed in (PFA) (#1004965000, Merck Millipore) at RT for 10 min and permeabilized overnight on ice with 70% EtOH. For γH2AX staining, cell cycle profile analysis, organoids were washed once with 10 mL PBS + 1% BSA and 0.02% tween, incubated with Phospho-Histone H2A.X (Ser 139)-Alexa Fluor 488 (eBioscience, #53-9865-82) for 30 min at RT, covered from light. For cell cycle profiling, organoids were incubated with RNAse (100 µg/ml) in PBS for 20 min at RT and with DAPI for 2 h, on ice, covered from light. For EdU detection, cells were stained by the Click-iT Plus EdU Pacific Blue Flow cytometry assay kit (Thermo fisher, #C10636) according to manufacturer's instructions. Flow cytometry was performed by using a BD FACS Celesta #660345.

**Immunofluorescence and live imaging.** Organoids were washed once in ice-cold-PBS, were collected in 1 mL ice-cold cell recovery solution (Corning, #354253) and 1 mL of ice-cold PBS in a 15 mL tube, and were incubated on ice for 10 minutes. Organoids were washed once with ice-cold PBS and were fixed by 4% PFA (#1004965000, Merck Millipore) for 20 min at RT and stored in PBS at 4 °C. For staining, organoids were transferred into 1.5 mL Eppendorf tubes. Organoids were permeabilized with PBS buffer containing 10% DMSO, 2% Triton X-100, and 10 g l$^{-1}$ BSA for 4 h at 4 °C. Organoids were stained with primary antibodies (γH2AX 1:400, Sigma, #05-636, and Ki67 1:200, Abcam, #ab15580) overnight, Alexa fluorophore-conjugated secondary antibodies (Invitrogen) for 4 h with DAPI for 1 h at 4 °C. Imaging was performed using a SP8 confocal microscope (Leica Microsystems). Light microscopy was performed using EVOS M5000 imaging system (Invitrogen).

*Image analysis immunofluorescence.* Image analysis was performed in ImageJ. For image analysis, images were converted into 32-bit images. Nuclei were automatically detected by the Stardist-2D macro based on the DAPI staining. Within these ROIs, intensities of gH2AX, KI67 or STAR were determined. STAR- and STAR$^+$ cells were identified as the nuclei with respectively 20% lowest and highest STAR intensities per organoid. For Ki67 positivity, a cutoff of intensity 20 was used.

**Table 2 Expansion medium composition for tumor progression organoids and patient-derived organoids (PDOs).**

|  | WT | AK | AP | APK | APKS | PDOs |
|---|---|---|---|---|---|---|
| 0.25 nM Wnt surrogate FC fusion protein (U-protein express BV, #N001) | x | – | – | – | – | – |
| R-spondin1-CM (homemade) | 20% v/v | – | – | – | – | 10% v/v |
| 10% v/v Noggin-CM (homemade) | x | x | x | x | – | x |
| 1x B27 (Fisher Scientific, #15360284) | x | x | x | x | x | x |
| 10 mM Nicotinamide (Sigma, A9165) | x | x | x | x | x | x |
| 1.25 mM N-acetyl-cysteine (Sigma, #N0636) | x | x | x | x | x | x |
| 3 µM SB202190 (Gentaur, #607-A1632) | x | x | x | x | x | x |
| 500 nM A83 (Biotechne, #2939) | x | x | x | x | x | x |
| 50 ng/mL EGF (Peprotech, #AF-100-15) | x | – | x | – | – | x |

**Table 3 Differentiation medium composition for tumor progression organoids and patient-derived organoids (PDOs).**

|  | WT | AK | AP | APK | APKS | PDOs |
|---|---|---|---|---|---|---|
| 0.125 nM Wnt surrogate FC fusion protein (U-protein express BV, #N001) | x | – | – | – | – | – |
| R-spondin-CM (homemade) | 10% v/v | – | – | – | – | 5% v/v |
| 10% v/v Noggin-CM (homemade) | x | x | x | x | – | – |
| 1x B27 (Fisher Scientific, #15360284) | x | x | x | x | x | – |
| 1.25 mM *N*-acetyl-cysteine (Sigma, #N0636) | x | x | x | x | x | – |
| 500 nM A83 (Biotechne, #2939)) | x | x | x | x | x | – |
| 50 ng/mL EGF (Peprotech, #AF-100-15) | x | – | x | – | – | – |

*Image analysis glucose sensor*. Imaging of the FLII12Pglu-700uDelta6 glucose FRET sensor was performed in 4 independent experiments. Data analysis was performed in ImageJ. Images of YFP and CFP channels were converted into 32-bit images, automatic thresholding was performed and YFP/CFP ratio was visualized by using the image calculator tool. YFP/YFP ratios were quantified by measuring the mean of the ratios in whole organoids.

**Growth curves**. 7 days after trypsinization organoids were replated into a 96-well plate (5 µL BME/well, 5 wells/condition). Upon 3 days of incubation in differentiation medium (t = 0), 5-FU was administered and organoids were imaged every 2 days at the EVOS M5000 imaging system. Organoid size was analyzed by manual analysis in ImageJ.

**Cell viability imaging**. 7 days after trypsinization, organoids were replated into a 96-well plate (5 µL Matrigel/well, 3 wells/condition). For cell viability analysis, organoid were stained with CyQUANT Direct assay (Thermofisher, #C35011) according to the manufacturer's instructions to detect the alive cells and Hoechst (#H1399, Life Technologies) to detect all dead and alive cells, for 1 h at 37 °C. Organoids were imaged at a Cell observer Z1 (Zeiss).

*Image analysis*. In imageJ, images were converted into 32-bit, and maximal projections of C1 (CyQUANT) and C3 (Hoechst) images were created. Organoids in both channels were automatically detected by the Stardist-2D macro to generate regions of interest (ROIs) reflecting respectively the alive parts (alive ROIs) and the total organoids (total ROIs). In the C1 images, the alive ROIs were masked and converted into binary images. Now, in the binary C1 images, the total ROIs were imported, and the %area of alive ROIs was determined within these total ROIs and used as a viability score per organoid.

**Outgrowth experiments and imaging**. 7 days after trypsinization, organoids were replated into a 24-well plate (4 droplets of 11 µL) and cultured on differentiation medium. Upon 3 days, medium was refreshed and DCA (20mM) was added. After 20 h, 5-FU (50 µM) was administrated and organoids were cultured for 48 h. Then drugs were washed away and organoids were cultured for another 7 days on differentiation medium. After 7 days, organoids were trypsinized into small clumps and replated in Matrigel in a 24-well plate and in a 96-well plate (5 µL Matrigel/well, 4 wells/condition). Upon 4 days, organoids in 24-well plate were imaged by EVOS. Organoids in 96-well plate were stained with CyQUANT Direct assay (Thermofisher, #C35011) according to the manufacturer's instructions and Hoechst for 1 h at 37 °C. Organoids were imaged at a Cell observer Z1 (Zeiss).

*Image analysis*. In imageJ, images were converted into 32-bit, and maximal projections of C1 (CyQUANT) images were created. Alive organoid (particles) were automatically detected by the Stardist-2D macro.

**Seahorse XF Flux analysis**. Seahorse Bioscience XFe24 Analyzer was used to measure extracellular acidification rates (ECAR) in milli pH (mpH) per min and oxygen consumption rates (OCR) in pmol O2 per min as previously described[67]. In short, organoids were seeded in 3 µL Matrigel per well in XF24 cell culture microplates (Seahorse Bioscience). 1 h before the measurements, culture medium was replaced by Assay medium and the organoids were incubated for 60 min at 37 °C. For experiments with DCA, DCA was also added to the Assay medium. For the mitochondrial stress test, culture medium was replaced by Seahorse XF Base medium (Seahorse Bioscience), supplemented with 10 mM glucose (Sigma-Aldrich), 2 mM L-glutamine (Sigma-Aldrich), 5 mM pyruvate (Sigma-Aldrich) and 0.56 µL ml$^{-1}$ NaOH (1M). During the test, 5 µM oligomycin, 2 µM FCCP and 1 µM of rotenone and antimycin A (all Sigma-Aldrich) were injected to each well after 18, 45, and 63 min, respectively. For the glycolysis stress test, culture medium was replaced by Seahorse XF Base medium, supplemented with 2 mM L-glutamine and 0.52 µL mL$^{-1}$ NaOH (1 M). During the test 10 mM glucose, 5 µM oligomycin and 100 mM 2-deoxyglucose (Sigma-Aldrich) were injected to each well after 18, 36, and 65 min, respectively. After injections, measurements of 2 min were performed in triplo, preceded by 4 min of mixture time. The first measurements after

oligomycin injections were preceded by 5 min mixture time, followed by 8 min waiting time for the mitochondrial stress test and 5 min mixture time followed by 10 min waiting time for the glycolysis stress test. OCR and ECAR values per group were normalized to the total amount of DNA present in all wells of the according group.

**Metabolomics**

*Materials*. Organic solvents were ULC-MS grade and purchased from Biosolve (Valkenswaard, The Netherlands). Chemicals and standards were analytical grade and purchased from Sigma-Aldrich (Zwijndrecht, The Netherlands). Water was obtained on the day of use from a Milli Q instrument (Merck Millipore, Amsterdam, The Netherlands).

*Sample preparation and LC-MS analysis*. For metabolomics, 3 wells with 200 µL of organoid-containing Matrigel were used per condition. During treatment, medium and drugs were refreshed 7 h before collection. Upon collection time, from each well, 500 µl medium per well was collected, pooled with medium from the other wells from the same condition and snap-frozen in liquid nitrogen in 2 mL Eppendorf tubes. Organoids from the same condition were pooled during collection. Organoids were washed once with ice-cold PBS, and subsequently collected in ice-cold PBS in a 15 mL falcon tube. Upon another wash with ice-cold PBS, organoids were transferred to 2 mL Eppendorf tubes, centrifuged, resuspended in 80% ice-cold methanol and snap-frozen in liquid nitrogen.

Samples were evaporated to dryness in a Labconco Centrivap (VWR, Amsterdam, The Netherlands). To the residue 350 µL water, 10 µL 1 mM ribitol internal standard in water, 375 µL methanol and 750 µL chloroform were added. After pulse vortex mixing, the samples were incubated for 2 h in a VWR thermostated shaker (900 rpm, 37 °C). After centrifugation at room temperature (10 min, 15,000 × *g*) the upper aqueous phase was quantitatively transferred to a clean 1.5 µL Eppendorf tube and evaporated to dryness overnight in the Labconco Centrivap. The residue was dissolved in 100 µL water, transferred to an injection vial and kept at 6 °C during LC-MS analysis.

The LC-MS analysis was performed using a 2.1 × 100 mm Atlantis premier BEH-C18 AX column (2.1 × 100, 2.5 µm) connected to a VanGuard column, both purchased from Waters (Etten-Leur, The Netherlands). The column setup was installed into an Ultimate 3000 LC system (Thermo Scientific, Breda, The Netherlands). The column outlet was coupled to a Thermo Scientific Q-Exactive FT mass spectrometer equipped with an HESI ion source. The UPLC system was operated at a flow rate of 250 µL min$^{-1}$ and the column was kept at 30 °C. The mobile phases consisted of 10 mM ammonium acetate and 0.04(v/v) ammonium hydroxide in water, pH9 (A), and acetonitrile (B), respectively. Upon 5 µL sample injection the system was kept at 0% B for 1 min followed by a 4 min linear gradient of 0–30% B. Thereafter, the gradient increased linearly to 95% B in 3 min and kept at 95% for 2 min. The column was regenerated at 0% B for 6 min prior to a next injection. All samples were injected three times (3 technical replicates). Mass spectrometry data were acquired over a scan range of m/z 72 to 900. The system was operated at −2.5 kV (negative mode) and 120,000 mass resolution. Further source settings were: transfer tube and vaporizer temperature 350 °C and 300 °C, and sheath gas and auxiliary gas pressure at 35 and 10, respectively. For high mass accuracy mass calibration was performed before each experiment. Raw data files were processed and analyzed using XCalibur Quan software.

**Statistics and reproducibility**. Statistical analysis for image analysis, flow cytometry and metabolomics results was performed by using Graphpad Prism 8. First Gaussian distribution of data was tested by a Shapiro-Wilk test to next apply parametric or non-parametric statistics. Details of statistics are described in the figure legends.

Sample sizes depicted in the figure legends refer to the number of independent experiments, unless stated differently. When the sample size refers to technical replicates, this is explained in the figure legends. For metabolomics, technical replicates came from repeated measurements. In all other cases, technical replicates came from distinct samples.

**Reporting summary**. Further information on research design is available in the Nature Research Reporting Summary linked to this article.

## Data availability

All data generated during this study are included in this article. Source data is provided as Supplementary Data 1 (main figures) and 2 (Supplementary figures). Uncropped and unedited blot images are included in Supplementary Fig. 7.

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

## Acknowledgements
We thank H.J.G. Snippert (UMC Utrecht) for sharing the stem cell activity reporter plasmid; S.E.M. van der Horst and H.J.G. Snippert (UMC Utrecht) for sharinig the *P53* overexpression construct; W. Frommer (Heinrich-Heine-University) for sharing the glucose FRET sensor plasmid; I. Verlaan (UMC Utrecht) for preparing R-spondin- and Noggin-conditioned medium; and A. Janssen and J. Lehman for critical reading of the manuscript. This work was financially supported by Dutch Cancer Society (KWF 2016-I 10471, KWF 2017-II 11315).

## Author contributions
Conceptualization, M.J.R.C., M.C.L., and B.M.T.B.; methodology and investigation, M.J.R.C., M.C.L., M.M., S.G., N.T.B.N., and S.K.S.R; metabolomics: M.C.G and E.C.A.S; development tumor progression organoid model: J.D. and H.C.; manuscript writing, M.J.R.C. and M.C.L.; review and editing, M.J.R.C. and B.M.T.B.; funding acquisition, M.J.R.C. and B.M.T.B.

## Competing interests
The authors declare no competing interests.
