## [Peer Review File · Communications Biology]

Reviewers' comments:

Reviewer #1 (Remarks to the Author):

Ludikhuizen et al. in the current manuscript "Rewiring glucose metabolism improves 5-FU efficacy in glycolytic p53-deficient colorectal tumors", report p53 deficiency induces pyrimidine imbalance which leads to DNA damage and cell death in human CRC organoids treated with 5-FU. Moreover, 5-FU toxicity can be enhanced in p53 deficient organoids, by rewiring glucose metabolism through inhibiting PDK using DCA. The empirical evidence supporting the reported findings is based entirely on in vitro experiments and in vivo experiments are needed to strengthen their observations. Authors need to address following concerns in order to be considered for publication at Communications Biology.

Major comments:

1. In vivo experiments demonstrating DCA or rewiring glucose metabolism improves 5-FU efficacy is needed, at least using APK or APKS models.
2. Rescue experiments showing dependency on p53 are needed in support of their claims.
3. Relative growth rates of AK, AP, APK and APKS normalized to WT organoids (relative size, number) is needed to appreciate the effects of p53 loss on 5-FU efficacy.
4. Concomitant changes in OCR upon DCA treatment with and without 5-FU treatment is needed.
5. Please clarify (Fig 2E&F) for how long Palbociclib was administered to the organoids and show cell cycle status under these conditions.

Minor Points:

1. Table highlighting percentages of cells in different cell cycle phases in WT, AK, AP, APK and APKS organoids and changes relative to WT upon 5-FU and DCA treatment will greatly emphasize the authors findings.
2. Fig 1C, western-blot and flow cytometry analysis in APK condition are not in line with each other.
3. Fig S3F, please include better representative images.
4. In Fig 2C and Fig 4D, under similar 5-FU treatment conditions - cell cycle profiles are significantly different. Please explain this discrepancy.
5. Better representation of organoid images and corresponding cell viability is needed.

Reviewer #2 (Remarks to the Author):

In the manuscript "Rewiring glucose metabolism improves 5-FU efficacy in glycolytic p53-deficient colorectal tumors" the authors addressed the resistance of colorectal tumors (CRC) to chemotherapy by 5-FU (5-fluorouracil), the major chemotherapy for CRC patients, 50% of whom are resistant to it. The authors used previously developed tumor progression organoid model (TPO) (Sequential cancer mutations in cultured human intestinal stem cells, Drost et al, 2015), where organoids were developed from healthy, normal colon and were subsequently genetically modified to generate the four most common driver mutations, APC^{KO}, KRAS^{G12D}, p53^{KO}, and SMAD4^{KO}. Their results suggest that in p53 null background (common in non-responders to 5-FU), 5-FU induces pyrimidine imbalance, which leads to DNA damage and cell death. In contrast, active p53 protects from these effects through inducing cell cycle arrest. Furthermore, they suggest that by rewiring glucose metabolism using DCA, the 5-FU toxicity can be enhanced by altering the nucleotide pool and without increasing toxicity in non-transformed cells.

I think that this manuscript addresses an important question and that the topic is relevant to the broader public and merits publication in the journal. I would suggest a minor revision before accepting it for publication in *Communication Biology*.

The authors need to provide additional/stronger evidence to support their conclusions:

1. Additional evidence should be provided that glycolysis inhibition promotes the 5-FU effect by using an alternative approach to DCA, for example, other inhibitors or a genetic alternation.
2. In addition to TPO, an alternative model should be used, e.g., patient organoids, organoids from another tumor type, cell line and xenograft model, animal genetic model....
3. It would be also interesting to see if there is a genetic signature in p53 KO tumors from patients suggesting a specific/glycolytic metabolic signature.
4. The authors should discuss why the effects of DCA are seen in APKS and APS organoids, but not in the AP model, which is p53 null.
5. The effect of DCA on the organoid size is very small (Fig. 6A). In addition to % live cells/organoid, the size of the organoid should be also reported and analyzed.
6. Authors should discuss the effects in p53 WT organoids (e.g., Dichloroacetate restores colorectal cancer chemosensitivity through the p53/miR-149-3p/PDK2-mediated glucose metabolic pathway (Liang et al, *Oncogene* 2020)), as the organoids are much smaller upon treatment – different mechanism.
7. Authors should report metabolites analyzed and comment on any other metabolic changes found.

Reviewer #3 (Remarks to the Author):

In this manuscript, authors show that 5-FU, an important anti-metabolite component of the chemotherapy cocktail used to treat colorectal cancer (CRC), induces DNA damage and cell death specifically in p53-deficient CRC organoids. They show that wild type (wt) p53 protects organoids from 5-FU-induced DNA damage and cell death by inducing the expression of p21, leading to G1 arrest. It was also demonstrated that 5-FU induces DNA damage in actively proliferating cell populations as it induces DNA damage during replication. 5-FU was shown to do so as a result of pyrimidine imbalance via rewiring glucose metabolism. Finally, authors showed that targeting Warburg effect by DCA potentiates 5-FU by altering the nucleotide pool.

This is a very well written manuscript with strong data supporting the claims. This manuscript contributes to our understanding of how an important chemotherapy agent, 5-FU works at mechanistic level. Figures are of high quality, and data is clearly presented. I have following points to improve it further:

1. The manuscript very well characterizes 5-FU response in different stages of CRC progression using the state-of-the-art organoid models. The contribution of p53 to mechanism of 5-FU-induced DNA damage and cell death is clearly shown (Figures 1, 2 and 3). However, the contribution of p53 to pyrimidine imbalance and glucose metabolism (Figures 4,5 and 6) is not clear and needs to be addressed.
2. An experiment with adding back wt p53 in AP, APK and APKS models with an output of examining DNA damage and metabolic alterations would further strengthen the manuscript. Furthermore, in patients, instead of p53 deletion, p53 mutation is very common. How does mutant p53 impact 5-FU response and does the mechanism proposed here hold true in that setting?
3. As the authors mentioned, the mechanism of 5-FU-mediated cell killing could be due to DNA damage or RNA toxicity depending on the dose and duration of the treatment. Authors used 100 microM in most of their experiments. How do authors justify this dose? A dose-response experiment, testing DNA damage, apoptosis and viability simultaneously, might help to define the dose range where the proposed mechanism (here: DNA damage-focused) in this work holds true and helps define its relevance to treatment setting in patients (discussed in the Discussion section).
4. Authors clearly state in the discussion section that "cellular behavior rather than being a specific cell type determines 5-FU sensitivity". And their data support this argument. However, in section 3 of the Results, they also state that "5-FU induces DNA damage in proliferating cancer (stem) cells". I would soften this statement, maybe by removing "stem" cell and just leave it as

proliferating cells as it is not clearly shown that those proliferating cells are CSCs and this argument cannot be generalized to other CSCs, e.g., cycling/quiescent stem cells.

5. The effect of DCA to potentiate the effect of 5-FU may be generalized to other chemotherapy agents which targets DNA replication considering the mechanism proposed. It would be very useful to test if DCA improves the effects of other chemo agents targeting replication.

Response to the reviewers

We appreciate the reviewers' comments. Please find below a detailed response to their suggestions.

Reviewer 1

Major comments:

1. In vivo experiments demonstrating DCA or rewiring glucose metabolism improves 5-FU efficacy is needed, at least using APK or APKS models.

We appreciate the suggestion of the reviewer. However, considering time limitations and upon suggestion of the other reviewers and the editor, we have chosen to include experiments with the colorectal cancer patient-derived organoids (PDOs) as an additional model system. These organoids have been previously sequenced and characterized and closely recapitulate the biology of colorectal tumors in the patients (van de Wetering et al., 2015). The results of these experiments are presented in figures 4, S4, 6, S5, and S6. In line with the tumor progression organoid (TPO) model, we observed DNA damage, S-phase accumulation and cell death upon 5-FU treatment in patient-derived organoids with non-functional p53. In PDO lines with functional p53, 5-FU induces P21 expression, G1 arrest and no DNA damage accumulation. However, differently from TPOs, these PDOs rapidly undergo apoptosis within 24-28 hours of 5-FU treatment independently of DNA damage. This indicates that in presence additional tumor-intrinsic factors, such as the presence of other oncogenic mutations, in addition to inducing a G1-arrest, p53 can induce DNA damage independent apoptosis. Moreover, we observed that DCA can also improve 5-FU efficacy in P19bt, a PDO-line with a glycolytic phenotype and non-functional p53. These observations are included, interpreted and discussed in the revised version of the manuscript (MS).

2. Rescue experiments showing dependency on p53 are needed in support of their claims.

We have introduced a doxycycline-inducible p53 overexpression (OE) system in AP, APK and APKS organoid lines. The results with the APK and APKS p53 OE lines are included in the revised manuscript in Figure 3 and S3 and in the results section (lines 162-170). In short, in agreement with our previous findings, we observed that upon overexpression of p53^{WT}, 5-FU induces p53 stabilization and induction of p21 expression in APK and APKS organoids (Figure 3G). Moreover, p53 addback prevented S-phase accumulation and DNA damage upon 5-FU treatment in these lines (Figure 3G, H). p53 overexpression also partially rescued cell death upon 5-FU treatment in these lines (Figure 3I, J). Unfortunately, we lost the AP-p53 OE line before we could finish all experiments. The (preliminary) results with this line however, also support the proposed mechanism and are presented in annex figure 1. P53 OE in the AP organoid line also prevented DNA damage and S/G2 phase accumulation upon 5FU treatment (Annex figure 1A-C). Titration of doxycycline concentration for this line would be required to increase the window of P21 expression between 5-FU-treated and non-treated AP-p53OE organoids.

Additionally, we performed p53^{WT}-rescue experiments in PDOs with non-functional p53, P16t and P19bt. In line with the TPO model, we found that reintroduction of p53^{WT} caused p21 expression and prevented DNA damage and S-phase accumulation upon 5-FU treatment. Furthermore, we observed a clear rescue of 5-FU-induced cell death in P19bt organoids, and a milder effect in p16t organoids, indicating that depending on certain tumor-intrinsic factors, such as the presence of other oncogenic mutations, in

addition to inducing a G1-arrest, p53 can alternatively induce DNA damage independent apoptosis. These observations are included in the revised manuscript in lines 184-191 and Figure 4 and S4.

3. Relative growth rates of AK, AP, APK and APKS normalized to WT organoids (relative size, number is needed to appreciate the effects of p53 loss on 5-FU efficacy).

In the current version of the manuscript, we have included growth curves for the untreated WT and the different tumor progression organoid lines (lines 94-97) (Figure S1A). Timepoint 0 represents the time point where 5-FU was added. These growth curves show that non-functional p53 and increasing number of mutations positively correlate with faster proliferation rates. Upon 5-FU treatment, all lines show reduced growth when compared to the non-treated controls (Figure S1B, C and annex figure 2A). We included the growth curves for 5-FU-treated WT and AK organoids in our manuscript to stress the cell cycle arrest upon 5-FU treatment in these lines. We do not include the growth curves of 5-FU-treated AP, APK and APKS organoids in our manuscript (Annex figure 2A), as in these lines the most prominent phenotype is reduced viability, and because the observed reduced growth phenotype is technically not independent of the reduced viability.

4. Concomitant changes in OCR upon DCA treatment with and without 5-FU treatment is needed.

In response to this comment, we have included OCR measurements upon DCA treatment in the revised manuscript (lines 239-241, Figure S5F,G). Upon DCA treatment, we found an increase in maximal respiration in APK and APKS organoids and an increase in basal respiration in APKS organoids. This is in line with the inhibition of pyruvate dehydrogenase kinase (PDK) and the consequent dephosphorylation of PDH (Figure S5D). These bioenergetic parameters were measured following a 24-hour DCA treatment, which is the relevant time point to address the metabolic state at the moment that 5-FU is added in the experiments (Annex figure 3A). As in this manuscript we do not focus on determining the metabolic changes upon 5-FU treatment, we do not include OCR measurements after 5-FU treatment in our manuscript. These analyses were nonetheless performed and results are presented in Annex figure 3B for reviewer's consideration. No significant changes in basal and maximal OCR were observed upon 5-FU or the 5-FU/DCA combination treatment.

5. Please clarify (Fig 2E&F) for how long Palbociclib was administered to the organoids and show cell cycle status under these conditions.

We appreciate the comment of the reviewer. We have added a scheme in the revised MS explaining the experimental set-up of the palbociclib experiments (Figure S3A). Furthermore, we included additional flow cytometry analysis that palbociclib induces a G1 arrest after 24 hours and that this arrest is sustained along the entire experiment (Figure S3B).

Minor Points:

1. Table highlighting percentages of cells in different cell cycle phases in WT, AK, AP, APK and APKS

organoids and changes relative to WT upon 5-FU and DCA treatment will greatly emphasize the authors findings.

In the revised MS, we included a table (Table 1) where we indicated the changes upon 5-FU to emphasize our findings (increases and decreases in green and red, respectively). We did not perform cell cycle profile analysis upon DCA treatment, as we performed EdU-incorporation analysis here (Figure 6G), which already shows that DCA does not influence proliferation rates and provides a deeper insight in proliferation status than cell cycle profiles.

2. Fig 1C, western-blot and flow cytometry analysis in APK condition are not in line with each other.

Agreed, we have now included a more representative western blot (representative of 5 independent experiments) (Figure 1C).

3. Fig S3F, please include better representative images.

Agreed, we have now included a better representative image. Besides, we performed nuclear segmentation and pin-pointed with dashed lines the cells with low to no γ H2AX staining in all channels. These cells show low signal in the STAR channel. In the revised MS this is referred to in figure S2F.

4. In Fig 2C and Fig 4D, under similar 5-FU treatment conditions - cell cycle profiles are significantly different. Please explain this discrepancy.

We thank the reviewer for bringing up this discrepancy. We reanalyzed our gating strategy to assure consistent gating across different experiments. The updated figures are now more comparable. However, we do still find a mild increase in G2-phase upon 5-FU treatment in figure 5D, whereas this was not found in earlier experiments represented in Figure 3C. Although we cannot be certain about the reason for this, we speculate that this could be possibly related to different BME/Matrigel batches used at earlier or later time points during the development of this project. Changes in organoid growth conditions seem to sometimes have (subtle) consequences on phenotypes, which has been noticed by us and others. Despite these small differences, the increase in S/G2 cells upon 5-FU is consistently found in all experiments.

5. Better representation of organoid images and corresponding cell viability is needed.

We revised these figures and in the current MS we provide zoomed-in images showing DIC, CYQUANT (alive), Hoechst (total) and the merge for both CYQUANT and Hoechst channels (Figure 7B, S6C).

Reviewer #2 (Remarks to the Author):

In the manuscript “Rewiring glucose metabolism improves 5-FU efficacy in glycolytic p53-deficient colorectal tumors” the authors addressed the resistance of colorectal tumors (CRC) to chemotherapy by 5-FU (5-fluorouracil), the major chemotherapy for CRC patients, 50% of whom are resistant to it. The authors used previously developed tumor progression organoid model (TPO) (Sequential cancer mutations in cultured human intestinal stem cells, Drost et al, 2015), where organoids were developed from healthy, normal colon and were subsequently genetically modified to generate the four most common driver mutations, APCKO, KRASG12D, p53KO, and SMAD4KO. Their results suggest that in p53 null background (common in non-responders to 5-FU), 5-FU induces pyrimidine imbalance, which leads to DNA damage and cell death. In contrast, active p53 protects from these effects through inducing cell cycle arrest. Furthermore, they suggest that by rewiring glucose metabolism using DCA, the 5-FU toxicity can be enhanced by altering the nucleotide pool and without increasing toxicity in non-transformed cells. I think that this manuscript addresses an important question and that the topic is relevant to the broader public and merits publication in the journal. I would suggest a minor revision before accepting it for publication in *Communication Biology*. The authors need to provide additional/stronger evidence to support their conclusions:

1. Additional evidence should be provided that glycolysis inhibition promotes the 5-FU effect by using an alternative approach to DCA, for example, other inhibitors or a genetic alternation.

We appreciate the comment of the reviewer. In the current study, we showed that DCA’s additive effect on 5-FU efficacy is rescued by TEPP-46 treatment (Figure 6I-K, S50,P). TEPP-46 is an activator of PKM leading to increased glycolysis (Figure 6A). Thus we conclude here that the additive effect of DCA on 5-FU is indeed due to its effect on the inhibition of glycolysis.

Nevertheless, we attempted to evaluate the effect of additional glycolytic inhibitors. We selected the glycolytic inhibitors 3-(3-Pyridinyl)-1-(4-pyridinyl)-2-propen-1-one (3-PO) and PFK-158, which are thought to inhibit PFKFB3 (Burmistrova et al., 2019; Clem et al., 2008; Emini Veseli et al., 2020). However, in analyses, 3-PO and PFK-158 increased glycolysis rather than decreasing it (Annex figure 4A, B). Therefore, we stopped further analysis with these compounds. Additionally, we tested the MCT-1 inhibitor SR13800. As shown in Annex figure 4C, 0.5 μ M SR13800 was sufficient to reduce glycolytic rates in APK organoids. Interestingly, this concentration also mildly, but significantly increased both 5-FU induced DNA damage and cell death (Annex figure 4D-F). These preliminary results suggests that, similarly to DCA, other compounds that inhibit glycolysis are able to improve 5-FU efficacy. We, however, do not to include these results in our manuscript as, due to time limitations, we were unable to optimize concentrations and timings to further characterize and validate this drug combination in all other organoid lines.

2. In addition to TPO, an alternative model should be used, e.g., patient organoids, organoids from another tumor type, cell line and xenograft model, animal genetic model....

Agreed. In the revised version of this MS, we have included experiments with the colorectal cancer patient-derived organoids (PDOs) as an additional model system. These organoids have been previously sequenced and characterized and closely recapitulating the biology of colorectal tumors in the patients

(van de Wetering *et al.*, 2015). The outcome of these experiments is presented in figures 4, S4, 6, S5, and S6. We selected different PDO lines and confirmed their previously determined p53 mutational status and their response to MDM2 inhibitor Nutlin-3 (Figure S4A). In line with the TPO model, we observed DNA damage, S-phase accumulation and cell death upon 5-FU treatment in patient-derived organoids with non-functional p53 (Figure 4B-E). Furthermore, in PDO lines with functional p53, we observed P21 expression, G1 arrest and no DNA damage accumulation. However, differently from TPOs, these PDOs rapidly undergo apoptosis within 24-28 hours of 5-FU treatment independently of DNA damage (Figure S4B-E). This indicates that, depending on certain tumor-intrinsic factors, such as the presence of other oncogenic mutations, in addition to inducing a G1-arrest, p53 can alternatively induce DNA damage independent apoptosis. These observations are included in the revised manuscript in lines 171-184 and figure 4 and S4.

Additionally, we tested the additive effect of DCA on 5-FU treated PDOs. We first analyzed the metabolic phenotype of the p53-deficient PDOs. While P9t and P19bt exhibit the Warburg effect, the glycolytic rate in P16t is similar to the wildtype colon organoids (Annex figure 5A, B and Figure S5J, K). In line with this, p16t did not show enhanced 5-FU-induced DNA damage upon DCA pretreatment (Figure S5M). In contrast, DCA does increase the 5-FU-induced DNA damage and cell death in the glycolytic P19bt organoid (figure 6H, S5L, S6A, B) and lines 246-250 and 265-271). In P9t, we did find an increase in DNA damage and cell death by DCA as a single treatment, indicating that for some tumors, DCA could already be beneficial for patients (Annex figure 5C-E). As we did not further study this, we did not include our findings for the P9t line in the manuscript. Related to additional models, we performed p53^{WT}-addback experiments in the TPO and two PDO lines P16t and P19bt. These results are included in the revised MS (Figure 4F-I, S4F, lines 184-191) and shortly explained in response to reviewer 1 (page 1, point 2)

2. It would be also interesting to see if there is a genetic signature in p53 KO tumors from patients suggesting a specific/glycolytic metabolic signature.

We agree with the reviewer that addressing the genetic signature of p53 KO tumors would be interesting. In general, p53^{WT} has been reported to slow down glycolysis and synthesis of nucleotides and lipids, and also to support mitochondrial maintenance and oxidative phosphorylation (reviewed in (Flöter *et al.*, 2017; Liu *et al.*, 2019; Vousden and Ryan, 2009)). In colorectal cancer cell lines, loss of p53 enhances glucose consumption and lactate production (Bao *et al.*, 2013; Jiang *et al.*, 2011). Nevertheless, our results in organoids show that the KRAS^{G12D} mutation, rather than the p53 loss of function, correlates with the metabolic change towards high glycolysis/Warburg effect. This is in line with previous studies, which are reviewed in (Kawada *et al.*, 2017) that we also refer to in the original and in the revised manuscript (lines 223-225). In the revised version, to emphasize this and prevent confusion, we re-ordered the bars in this bar graph to fit the same order as used in all other figures (Figure 6B). Moreover, we analyzed the bioenergetics upon addback of p53^{WT} in p53-deficient TPOs. We did not find significant changes in the glycolytic phenotype upon 24 hours of p53 addback, indicating that p53 does not have major effects on glycolysis in this model (Annex figure 1D). Therefore, rather than the importance of p53 in metabolic changes in CRC, we discuss constitutively active (mutant) KRAS in that respect.

3. The authors should discuss why the effects of DCA are seen in APKS and APS organoids, but not in the AP model, which is p53 null.

In line with our argumentation on the previous point, we observed an enhanced glycolytic phenotype in CRC organoids carrying the KRAS^{G12D} mutation, whereas p53^{KO} does not seem to be a discriminatory factor for the glycolytic phenotype in our organoids (Figure 6B). As the AP line does not carry a KRAS^{G12D} mutation and does not exhibit the Warburg effect, it is not expected that DCA will enhance 5-FU efficacy. We clarified this in the revised manuscript in line 244-246 and 248-250 .

5. The effect of DCA on the organoid size is very small (Fig. 6A). In addition to % live cells/organoid, the size of the organoid should be also reported and analyzed.

We agree with the reviewer that no major additive effect of DCA on the size of 5-FU-treated organoids can be observed. Quantification of the organoid area upon the different treatments shows only minor (10-20%), but significant effects of DCA on the size of 5-FU treated organoids (Annex figure 2B-D). We did not report the size of compromised organoids upon 5FU-treatment as the size can reflect both loss of proliferation and cell death. Moreover, fully compromised organoids can be larger than alive small alive organoids. Therefore, we still think that viability is a better parameter to assess drug efficacy than organoid size.

6. Authors should discuss the effects in p53 WT organoids (e.g., Dichloroacetate restores colorectal cancer chemosensitivity through the p53/miR-149-3p/PDK2-mediated glucose metabolic pathway (Liang et al, Oncogene 2020)), as the organoids are much smaller upon treatment – different mechanism.

In our manuscript, we show that DCA does not affect proliferation upon 24 hours of treatment. This does not rule out that prolonged exposure to DCA will affect proliferation and other phenotypes. In fact, metabolic transitions can regulate proliferation and differentiation processes in many different ways, which has been previously shown by us and others (Dahan et al., 2019; Ludikhuize and Rodríguez Colman, 2021; Wellen and Thompson, 2012). This could be underlying the observed phenotype of reduced tumor growth in xenograft models upon DCA and 5-FU and DCA combination treatments in Liang et al, Oncogene 2020 and the reduced size of our WT organoids upon DCA treatment. We have mentioned this in the discussion section of the revised MS (lines 343-345).

7. Authors should report metabolites analyzed and comment on any other metabolic changes found.

Our metabolomics analysis method was optimized to detect nucleotides, however we were able to detect additional metabolites. We include a table with the metabolites that were measured within a reliable range and that showed consistent changes across independent experiments (Annex Table 1 and 2). However, these results are to us not sufficient to draw any additional conclusions and therefore are not included in the revised MS. Related to any additional metabolic changes, we included OCR measurements to show the effect of DCA on mitochondrial respiration in the revised MS (Figure S5F, G). These analyses show that DCA increases mitochondrial respiration in APK and APKS organoids, further confirming the inhibition of PDK and the consequent activation of PDH (Figure S5D).

Reviewer #3 (Remarks to the Author):

In this manuscript, authors show that 5-FU, an important anti-metabolite component of the chemotherapy cocktail used to treat colorectal cancer (CRC), induces DNA damage and cell death specifically in p53-deficient CRC organoids. They show that wild type (wt) p53 protects organoids from 5-FU-induced DNA damage and cell death by inducing the expression of p21, leading to G1 arrest. It was also demonstrated that 5-FU induces DNA damage in actively proliferating cell populations as it induces DNA damage during replication. 5-FU was shown to do so as a result of pyrimidine imbalance via rewiring glucose metabolism. Finally, authors showed that targeting Warburg effect by DCA potentiates 5-FU by altering the nucleotide pool.

This is a very well written manuscript with strong data supporting the claims. This manuscript contributes to our understanding of how an important chemotherapy agent, 5-FU works at mechanistic level. Figures are of high quality, and data is clearly presented. I have following points to improve it further:

1. The manuscript very well characterizes 5-FU response in different stages of CRC progression using the state-of-the-art organoid models. The contribution of p53 to mechanism of 5-FU-induced DNA damage and cell death is clearly shown (Figures 1, 2 and 3). However, the contribution of p53 to pyrimidine imbalance and glucose metabolism (Figures 4,5 and 6) is not clear and needs to be addressed.

The reviewer requests more insights in the contribution of p53 to glucose metabolism and pyrimidine imbalance. Although interesting, it is not the aim of this study to describe the role of p53 on glucose and nucleotide metabolism. Importantly, we find that constitutively active KRAS (KRAS^{G12D} mutation), rather than non-functional p53, causes the metabolic change towards the Warburg effect (Figure 6B) in the organoid model system. In agreement with this conclusion, re-introduction of functional p53 in the AP, APK and APKS organoids did not result in significant changes in bioenergetics (Annex figure 1D). We now interpret reviewer's question to suggest that the differential response to 5-FU could be a consequence of the loss of p53 in decreasing the nucleotide pool. Metabolomics analysis in WT, AK, AP and APKS organoids revealed that organoids without p53 have increased TTP and dATP levels compared to p53^{WT} organoids (Annex figure 1E). Although other deoxynucleotides were unfortunately not detected, this indicates that the high sensitivity of p53-deficient organoids to 5-FU-induced DNA damage cannot be explained by a p53-induced effect on the nucleotide pool. Related to that, we observed that 5-FU in both WT and in p53-deficient organoids exerts its effect through inhibition of TS, as in both cases increased cellular and extracellular deoxyuridine levels are observed (Figure 5A, Annex fig 1F). Altogether this further supports that non-functional p53- induced sensitivity relies on the proposed mechanism rather than in a specific effect of p53 on the metabolism of glucose or of 5-FU.

2. An experiment with adding back wt p53 in AP, APK and APKS models with an output of examining DNA damage and metabolic alterations would further strengthen the manuscript. Furthermore, in patients, instead of p53 deletion, p53 mutation is very common. How does mutant p53 impact 5-FU response and does the mechanism proposed here hold true in that setting?

We thank the reviewer for this suggestion and agree that p53 addback experiments will strengthen the findings of our manuscript. A brief explanation of the outcome of those experiments can be found in response to R2 (first page, point 2) and in the revised MS in the results section and in Figure 3, S3, 4 and S4.

Related to the importance of p53 in metabolism, we do not find p53 a discriminating factor in the glycolytic phenotype of the organoids. In line with previous studies, we found that organoids carrying an oncogenic KRAS mutation (constitutively active) have higher glycolytic rates when compared to organoids without this mutation. To further clarify this point, we changed the order of this graph (Figure 6B) to fit the order of the other figures. To further confirm that p53 does not have a significant effect on bioenergetics, we run Seahorse experiments on the p53-addback lines AP, APK and APKS. These indeed showed that indeed, addback of p53 does not change the glycolytic phenotype of these organoids upon 24 hours of doxycycline treatment (Annex fig 1D).

Related to mutant p53, the PDO lines we selected carried different p53 mutations (van de Wetering *et al.*, 2015) (P16t: p.A161T and p.L194R: loss of function mutations, P9T: p.R175H: gain of mutation, P19bt: R273C: gain of function mutation and Q331_splice variant. Despite these mutations, these organoids were not responsive to Nutlin-3 and responded to 5-FU in a similar way as P53-deficient TPOs.

3. As the authors mentioned, the mechanism of 5-FU-mediated cell killing could be due to DNA damage or RNA toxicity depending on the dose and duration of the treatment. Authors used 100 microM in most of their experiments. How do authors justify this dose? A dose-response experiment, testing DNA damage, apoptosis and viability simultaneously, might help to define the dose range where the proposed mechanism (here: DNA damage-focused) in this work holds true and helps define its relevance to treatment setting in patients (discussed in the Discussion section).

For our study we were interested in the mechanisms underlying 5-FU-induced cell death. Based on literature, we selected a 5-FU concentration that elicit cell death. Others have shown that 100 μ M of 5-FU clearly induced cell death in HT-29 cells after 24 hours and that (Glazer and Lloyd, 1982). In line with this, 100 μ M of 5-FU was also enough to induce 90% cell death in of HCT-8 cells within 4 days (Humeniuk *et al.*, 2009). Thus, we analyzed APKS organoids response to different concentrations of 5-FU (ranging from 1 to 200 μ M). We analyzed the cultures after 48 hours of treatment and found compromised viability starting from 100 μ M and, importantly, this phenotype was absent in wild type colon organoids. Therefore, we decided on 100 μ M as the working 5-FU concentration. Nonetheless, upon the reviewer's request, we performed DNA damage analysis and cell cycle profiling in APKS organoids with lower 5-FU concentrations as reported in patients' plasma (8 μ M) (Zheng and Wang, 2005). As reported in Annex figure 6, preliminary data shows that 1 μ M, 5 μ M and 25 μ M of 5FU also induce DNA damage and S-phase accumulation. Furthermore, these concentrations reduced cell death after 6 days of 5-FU treatment. This indicates that our mechanism in a non-functional p53 scenario also holds true for lower 5-FU concentrations, albeit that effects on viability are observed only at later time points (2 versus 6 days).

4. Authors clearly state in the discussion section that "cellular behavior rather than being a specific cell type determines 5-FU sensitivity". And their data support this argument. However, in section 3 of the Results, they also state that "5-FU induces DNA damage in proliferating cancer (stem) cells". I would soften

this statement, maybe by removing “stem” cell and just leave it as proliferating cells as it is not clearly shown that those proliferating cells are CSCs and this argument cannot be generalized to other CSCs, e.g., cycling/quiescent stem cells.

We thank the reviewer for this suggestion and we removed the word “stem” in this sentence (line 120).

5. The effect of DCA to potentiate the effect of 5-FU may be generalized to other chemotherapy agents which targets DNA replication considering the mechanism proposed. It would be very useful to test if DCA improves the effects of other chemo agents targeting replication.

We agree that it would be interesting to find out whether the potentiating effect of DCA on 5-FU can be generalized to other replication-targeting therapeutic agents. However, we consider such a laborious study is out of the scope of this work. Nevertheless, we indeed include this point in the discussion of the revised MS (lines 348-352).

References:

Bao, Y., Mukai, K., Hishiki, T., Kubo, A., Ohmura, M., Sugiura, Y., Matsuura, T., Nagahata, Y., Hayakawa, N., Yamamoto, T., et al. (2013). Energy management by enhanced glycolysis in G1-phase in human colon cancer cells in vitro and in vivo. *Mol Cancer Res* 11, 973-985. 10.1158/1541-7786.MCR-12-0669-T.

Burmistrova, O., Olias-Arjona, A., Lapresa, R., Jimenez-Blasco, D., Eremeeva, T., Shishov, D., Romanov, S., Zakurdaeva, K., Almeida, A., Fedichev, P.O., and Bolaños, J.P. (2019). Targeting PFKFB3 alleviates cerebral ischemia-reperfusion injury in mice. *Sci Rep* 9. 10.1038/s41598-019-48196-z.

Clem, B., Telang, S., Clem, A., Yalcin, A., Meier, J., Simmons, A., Rasku, M.A., Arumugam, S., Dean, W.L., Eaton, J., et al. (2008). Small-molecule inhibition of 6-phosphofructo-2-kinase activity suppresses glycolytic flux and tumor growth. *Mol Cancer Ther* 7, 110-120. 10.1158/1535-7163.mct-07-0482.

Dahan, P., Lu, V., Nguyen, R.M.T., Kennedy, S.A.L., and Teitell, M.A. (2019). Metabolism in pluripotency: Both driver and passenger? *J Biol Chem* 294, 5420-5429. 10.1074/jbc.TM117.000832.

Emini Veseli, B., Perrotta, P., Van Wielendaele, P., Lambeir, A.M., Abdali, A., Bellosta, S., Monaco, G., Bultynck, G., Martinet, W., and De Meyer, G.R.Y. (2020). Small molecule 3PO inhibits glycolysis but does not bind to 6-phosphofructo-2-kinase/fructose-2,6-bisphosphatase-3 (PFKFB3). *FEBS Lett* 594, 3067-3075. 10.1002/1873-3468.13878.

Flöter, J., Kaymak, I., and Schulze, A. (2017). Regulation of Metabolic Activity by p53. *Metabolites* 7. 10.3390/metabo7020021.

Glazer, R.I., and Lloyd, L.S. (1982). Association of cell lethality with incorporation of 5-fluorouracil and 5-fluorouridine into nuclear RNA in human colon carcinoma cells in culture. *Mol Pharmacol* 21, 468-473.

Humeniuk, R., Menon, L.G., Mishra, P.J., Gorlick, R., Sowers, R., Rode, W., Pizzorno, G., Cheng, Y.-C., Kemeny, N., Bertino, J.R., and Banerjee, D. (2009). Decreased levels of UMP kinase as a mechanism of fluoropyrimidine resistance. *Mol Cancer Ther* 8, 1037-1044. 10.1158/1535-7163.MCT-08-0716.

Jiang, P., Du, W., Wang, X., Mancuso, A., Gao, X., Wu, M., and Yang, X. (2011). p53 regulates biosynthesis through direct inactivation of glucose-6-phosphate dehydrogenase. *Nat Cell Biol* *13*, 310-316. 10.1038/ncb2172.

Kawada, K., Toda, K., and Sakai, Y. (2017). Targeting metabolic reprogramming in KRAS-driven cancers. *Int J Clin Oncol* *22*, 651-659. 10.1007/s10147-017-1156-4.

Liu, J., Zhang, C., Hu, W., and Feng, Z. (2019). Tumor suppressor p53 and metabolism. *J Mol Cell Biol* *11*, 284-292. 10.1093/jmcb/mjy070.

Ludikhuize, M.C., and Rodríguez Colman, M.J. (2021). Metabolic Regulation of Stem Cells and Differentiation: A Forkhead Box O Transcription Factor Perspective. *Antioxidants & Redox Signaling* *34*, 1004-1024. 10.1089/ars.2020.8126.

van de Wetering, M., Francies, H.E., Francis, J.M., Bounova, G., Iorio, F., Pronk, A., van Houdt, W., van Gorp, J., Taylor-Weiner, A., Kester, L., et al. (2015). Prospective derivation of a living organoid biobank of colorectal cancer patients. *Cell* *161*, 933-945. 10.1016/j.cell.2015.03.053.

Vousden, K.H., and Ryan, K.M. (2009). p53 and metabolism. *Nat Rev Cancer* *9*, 691-700. 10.1038/nrc2715.

Wellen, K.E., and Thompson, C.B. (2012). A two-way street: reciprocal regulation of metabolism and signalling. *Nat Rev Mol Cell Biol* *13*, 270-276. 10.1038/nrm3305.

Zheng, J.-F., and Wang, H.-D. (2005). 5-Fluorouracil concentration in blood, liver and tumor tissues and apoptosis of tumor cells after preoperative oral 5'-deoxy-5-fluorouridine in patients with hepatocellular carcinoma. *World J Gastroenterol* *11*, 3944-3947. 10.3748/wjg.v11.i25.3944.

Annex figure 1. A) Western blot detection of p53, P21, γ H2AX and β -actin in doxycycline-inducible AP p53 OE organoids, treated with 5-FU for 48 hours (doxycycline treatment started 16 hours before 5-FU administration). **B, C)** Quantification of cells with DNA damage (B) and cell cycle profiling by flow cytometry (C) of doxycycline-inducible AP p53 OE organoids, treated with 5-FU for 48 hours (doxycycline treatment started 16 hours before 5-FU administration) and stained with anti- γ H2AX (B) or dapi (C) (mean \pm SEM, n = 2). **D)** Extracellular acidification rate (ECAR) of doxycycline-inducible p53 OE AP, APK and APKS organoids treated with doxycycline for 24 hours, determined by a glycolysis stress test by Seahorse XF analysis (mean \pm SEM, 3-4 technical replicates, one-way ANOVA, Sidak's multiple comparisons test). **E)** Detection of TTP and dATP by metabolomics in WT, AK, AP and APKS organoids (mean \pm SEM, 3 technical replicates). **F)** deoxyuridine and TTP detection by metabolomics in WT organoids treated with 5-FU for 30 hours (mean \pm SEM, 3 technical replicates, one-way ANOVA, Sidak's multiple comparisons test).

Annex figure 2. A) Growth curves of AP, APK and APKS organoids treated with and without 5-FU for 6 days, quantified from brightfield images. Dashed lines indicate that organoids showed loss of viability along the experiment (mean \pm SEM, two independent experiments). **B, C, D)** Area of WT, APK and APKS organoids treated with 5-FU for 7 days (DCA treatment started 20 hours before 5-FU administration). Area is quantified based on CYQUANT and Hoechst staining (images... and ...) (B: median, 237-720 organoids from 3 independent experiments, Kruskal-Wallis test, Dunn's multiple comparisons test, C,D: mean area relative to Ctrl (C) and (5-FU) conditions).

ns: non-significant, ** $p < 0.01$, *** $p < 0.001$, **** $p < 0.0001$.

B Seahorse XF 2:

Annex figure 3. A) Schematic overview of experimental set-up of experiments to determine DNA damage and viability upon DCA and 5FU treatments. Experimental read-outs indicated with MS are included in the manuscript **B)** Basal and maximal respiration of APK and APKS organoids treated with 5-FU and DCA (on day 2: Seahorse XF 2), determined by a Seahorse XF mitochondrial stress test (OCR is normalized to the non-treated controls, mean \pm SEM, n=5-6, one-way Anova, Sidak's multiple comparisons test). Ns: non-significant.

Annex figure 4. A-C) Extracellular acidification rate (ECAR) APK (A and C) and APKS (B) organoids treated with 3-PO (20 μ M), PFK-158 (20 μ M) or SR13800 (0.5 μ M) for 24 hours, determined by a glycolysis stress test by Seahorse XF analysis (mean \pm SEM, 4-5 technical replicates (A,B), n=3 independent experiments (C), unpaired t-test). **D)** Quantification of cells with DNA damage by flow cytometry APK organoids, treated with 5-FU for 48 hours (SR13800 treatment started 20 hours before 5-FU administration) and stained with anti- γ H2AX (mean \pm SEM, n = 3-5, one-way ANOVA, Sidak's multiple comparisons test). **E, F)** Representative brightfield images (E) and cell viability analysis (F) of APK organoids treated with 5-FU for 6 days. SR13800 treatment started 20 hours before 5-FU administration. (scale bar = 500 μ m, mean \pm SEM, n = 7-10, one-way ANOVA, unpaired t-test).

* $p < 0.05$, ** $p < 0.01$, **** $p < 0.0001$.

Annex figure 5. A) Extracellular acidification rate (ECAR) of WT, P16t, P9t and P19bt organoids determined by glycolysis stress test by Seahorse XF analysis (mean \pm SEM, n = 3-5, one-way ANOVA, Sidak's multiple comparisons test). **B)** Basal oxygen consumption rate (OCR) and maximal respiration of WT, P16t and P19bt organoids determined by mitochondrial stress test by Seahorse XF analysis (mean \pm SEM, n = 4-6, one-way ANOVA, Sidak's multiple comparisons test). **C)** Extracellular acidification rate (ECAR) of P9t organoids treated with 30 mM of DCA, determined by glycolysis stress test by Seahorse XF analysis (mean \pm SEM, n = 4-5 technical replicates, unpaired t-test, representative for two independent experiments). **D)** Quantification of cells with DNA damage by flow cytometry of P9t organoids treated with 5-FU for 48 hours and stained with anti- γ H2AX. DCA treatment (30 mM) was started 20 hours before 5-FU administration (mean \pm SEM, n = 4, one-way ANOVA, Sidak's multiple comparisons test). **E)** Cell viability analysis of P9t organoids treated with 5-FU for 6 days. DCA treatment (30 mM) was started 20 hours before 5-FU administration (mean \pm SEM, n = 5, one-way ANOVA, Sidak's multiple comparisons test).

Annex figure 6. A) Brightfield images of APKS organoids treated with 1, 5, 25, 100 or 200 μM 5-FU for 48 hours. **B,C)** Quantification of cells with DNA damage and cell cycle profiling (C) by flow cytometry of APKS organoids, treated with 1, 5 or 25 μM 5-FU for 48 hours and stained with anti- γH2AX (B) or dapi (C) (N=1). **D, E)** Brightfield images (D) and cell viability analysis (E) of APKS organoids treated with 1, 5 or 25 μM 5-FU for 6 days (scale bar = 300 μm , N = 1).

		CTRL		5FU	
		mean	sd	mean	sd
glycolysis	G6P+F6P	3.03E+08	4.66E+06	1.96E+08	1.77E+06
	DHAP	1.83E+07	3.86E+05	2.60E+06	3.68E+04
	PEP	2.59E+08	2.65E+06	1.99E+08	1.82E+06
PPP	gluconic acid	3.35E+06	5.97E+04	3.27E+06	4.19E+04
other	glutamine	3.05E+08	1.61E+06	3.43E+08	1.37E+06
nucleotides	ATP	6.96E+08	2.80E+07	5.61E+08	1.11E+07
	CTP	4.00E+07	7.30E+05	3.00E+07	6.10E+05
	dATP	ND		ND	
	GTP	1.14E+08	5.00E+06	7.92E+07	1.29E+06
	TTP	4.42E+06	8.87E+05	1.41E+06	2.93E+04
	TDP	2.79E+07	1.25E+06	8.98E+06	1.02E+06
	dUDP	ND		2.08E+06	2.06E+05
	dUMP	ND		1.18E+07	2.34E+05
	UTP	1.64E+08	3.66E+06	5.38E+07	1.22E+06
	deoxyuridine	1.19E+06	3.45E+03	3.23E+06	1.21E+05
	deoxyuridine media	3.94E+06	8.91E+04	4.08E+07	5.02E+05

Annex table 1. Detected metabolites by metabolomics of APKS organoids treated with 5-FU for 30 hours, normalized by protein content (AU/protein ($\mu\text{g}/\text{ml}$)).

		Experiment 1				Experiment 2							
		CTRL		DCA		CTRL		DCA		TEPP		TEPP DCA	
		mean	sd	mean	sd	mean	sd	mean	sd	mean	sd	mean	sd
Glycolysis	G6P+F6P	1.63E+08	6.12E+05	1.70E+08	3.49E+05	3.61E+07	7.28E+05	5.66E+07	2.64E+06	5.99E+07	1.50E+06	9.27E+07	6.00E+06
	DHAP	2.59E+06	1.24E+05	7.53E+06	3.97E+05	7.13E+05	1.18E+05	1.07E+06	3.04E+05	4.38E+06	6.76E+05	3.03E+06	2.66E+05
	PEP	1.09E+08	7.73E+05	1.64E+08	1.85E+06	6.69E+06	3.31E+05	7.69E+06	8.85E+05	4.29E+06	5.14E+05	1.82E+06	4.69E+05
PPP	gluconic acid	4.53E+06	1.47E+05	3.66E+06	1.49E+05	1.45E+06	1.83E+05	1.16E+06	3.39E+05	1.47E+06	4.83E+05	7.66E+05	2.09E+05
Other	glutamine	1.14E+08	3.09E+05	2.67E+08	1.61E+06	2.57E+07	1.55E+06	5.78E+07	2.20E+06	9.43E+06	4.38E+05	4.37E+07	2.20E+06
Nucleotides	ATP	9.36E+08	2.00E+07	4.98E+08	1.11E+07	5.99E+08	2.99E+07	4.31E+08	2.12E+07	8.81E+08	2.48E+07	6.54E+08	2.63E+07
	CTP	5.52E+07	9.93E+05	3.41E+07	9.46E+05	4.56E+06	4.52E+05	4.93E+06	8.16E+05	8.79E+06	7.77E+05	5.15E+06	8.62E+05
	dATP	ND		ND		3.21E+06	1.39E+05	1.89E+06	1.88E+05	5.86E+06	2.00E+05	2.96E+06	1.57E+05
	GTP	1.15E+08	3.36E+06	7.82E+07	1.60E+06	ND		ND		ND		ND	
	TTP	6.70E+06	2.03E+05	4.31E+06	1.53E+05	2.72E+06	6.69E+05	1.52E+06	3.17E+05	4.62E+06	8.91E+05	3.02E+06	7.83E+05
	UTP	2.25E+08	2.70E+06	1.27E+08	3.30E+06	8.53E+06	1.42E+06	1.02E+07	1.12E+06	1.71E+07	1.33E+06	9.50E+06	1.01E+06

Annex table 2. Detected metabolites by metabolomics of APKS organoids treated with DCA and TEPP-46 for 24 hours, normalized by protein content (AU/protein ($\mu\text{g/ml}$)).

REVIEWERS' COMMENTS:

Reviewer #1 (Remarks to the Author):

Thank you for making sincere efforts to address the raised concerns. With the additional experiments performed and incorporated changes, the revised manuscript is acceptable for publication.

Reviewer #2 (Remarks to the Author):

In the revised manuscript "Rewiring glucose metabolism improves 5-FU efficacy in glycolytic p53-deficient colorectal tumors" the authors sufficiently addressed most of the issues raised. The only thing I would recommend being more specific in conclusions or even in the title, stating that: "Rewiring glucose metabolism improves 5-FU efficacy in glycolytic p53-deficient/ KRASG12D colorectal tumors. Results (and previous evidence) suggest that the KRASG12D mutation, rather than the p53 loss of function, correlates with the metabolic change towards high glycolysis/Warburg effect, pointing toward a subgroup of tumors (organoids) that benefit from this combination of treatment.

Some minor suggestions:

1. Line75: AK – specify
2. Line 79: p53 or P53 – use consistently throughout the manuscript
3. Line 128: typo - growth
4. Fig. 3e and f do not correspond well

Reviewer #3 (Remarks to the Author):

Most of my concerns are addressed adequately, and the manuscript is further improved.

Dear editorial team,

We are glad to hear that you, in principle, accept the manuscript. We have addressed the minor suggestions of Reviewer 2 as follows:

R2: *The only thing I would recommend being more specific in conclusions or even in the title, stating that: "Rewiring glucose metabolism improves 5-FU efficacy in glycolytic p53-deficient/ KRASG12D colorectal tumors. Results (and previous evidence) suggest that the KRASG12D mutation, rather than the p53 loss of function, correlates with the metabolic change towards high glycolysis/Warburg effect, pointing toward a subgroup of tumors (organoids) that benefit from this combination of treatment.*

- In line with this comment we have added KRASG12D to the title and we have incorporated the concept in the abstract (line 27), introduction (line 83), results (line 224) and rephrased in the discussion (lines 340-343).

1. Line 75: AK – specify

- We have.

2. Line 79: p53 or P53 – use consistently throughout the manuscript

- We appreciate the comment. We have corrected this point. In the current version, when referring to the gene we consistently use capitals (P53) and lowercase when referring to the protein (p53).

3. Line 128: typo – growth

- Amended.

4. Fig. 3e and f do not correspond well

- *We have revised the figure and we do not find any inconsistencies. We are open to make amends if the comment could be clarified.*

Thank you for your positive response and we hope that you find the revised manuscript and submitted files suited for publication.

Kind Regards,

Maria Rodriguez Colman (corresponding author and on behalf of all authors).

Center for Molecular Medicine, dLAB

University Medical Center Utrecht

Universiteitsweg 100, 3584 CG Utrecht, The Netherlands

Phone: 31-88-75 68093

Email: m.j.rodriuezcolman@umcutrecht.nl